# Silencing of transposable elements may not be a major driver of regulatory evolution in primate iPSCs

Michelle C Ward[1,2]*, Siming Zhao[1], Kaixuan Luo[1], Bryan J Pavlovic[1], Mohammad M Karimi[3], Matthew Stephens[1,4], Yoav Gilad[1,2]*

[1]Department of Human Genetics, University of Chicago, Chicago, United States; [2]Department of Medicine, University of Chicago, Chicago, United States; [3]MRC London Institute of Medical Sciences, Imperial College, London, United Kingdom; [4]Department of Statistics, University of Chicago, Chicago, United States

**Abstract** Transposable elements (TEs) comprise almost half of primate genomes and their aberrant regulation can result in deleterious effects. In pluripotent stem cells, rapidly evolving KRAB-ZNF genes target TEs for silencing by H3K9me3. To investigate the evolution of TE silencing, we performed H3K9me3 ChIP-seq experiments in induced pluripotent stem cells from 10 human and 7 chimpanzee individuals. We identified four million orthologous TEs and found the SVA and ERV families to be marked most frequently by H3K9me3. We found little evidence of inter-species differences in TE silencing, with as many as 82% of putatively silenced TEs marked at similar levels in humans and chimpanzees. TEs that are preferentially silenced in one species are a similar age to those silenced in both species and are not more likely to be associated with expression divergence of nearby orthologous genes. Our data suggest limited species-specificity of TE silencing across 6 million years of primate evolution.
DOI: https://doi.org/10.7554/eLife.33084.001

*For correspondence:
mcward@uchicago.edu (MCW);
gilad@uchicago.edu (YG)

**Competing interests:** The authors declare that no competing interests exist.

## Introduction

Over half of primate genomes are annotated as transposable elements (TEs) (*Jurka, 2000*; *de Koning et al., 2011*). Primate TE sequences reflect their evolutionary history; some TEs are conserved over multiple phylogenetic clades and orders, whereas others are restricted to particular lineages. Considering the degree of sequence similarity between the human and chimpanzee genomes, it is not surprising that many of the same TEs are present in both species (*Chimpanzee Sequencing and Analysis Consortium, 2005*; *Ramsay et al., 2017*). In general, TE activity, especially from endogenous retroviruses (ERVs), has declined in the hominid lineage relative to other mammalian lineages (*Lander et al., 2001*). However, a fraction of evolutionarily recent TEs are active in the human genome, including HERV-K ERVs (*Tönjes et al., 1996*; *Medstrand and Mager, 1998*; *Fuchs et al., 2013*; *Klawitter et al., 2016*; *Wildschutte et al., 2016*), members of the Alu (*Batzer and Deininger, 1991*; *Batzer et al., 1991*), L1 (*Kazazian et al., 1988*, *Brouha et al., 2003*), and SVA (*Wang et al., 2005*; *Ostertag et al., 2003*) families. Most studies on mammalian TEs have focused on humans and mice, with a handful that describe the ERV and L1 TE families in chimpanzees (*Yohn et al., 2005*; *Mun et al., 2014*; *Marchetto et al., 2013*).

Because of their transposition competence, TEs are considered a potential source of regulatory innovation (*Kazazian, 2004*; *Cordaux and Batzer, 2009*). Indeed, TEs can serve as primate-specific regulatory sequence (*Jacques et al., 2013*; *Trizzino et al., 2017*), act as enhancer elements (*Lynch et al., 2011*; *Chuong et al., 2013*; *Xie et al., 2013*), carry transcription factor binding sites (*Wang et al., 2007*; *Bourque et al., 2008*; *Kunarso et al., 2010*; *Schmidt et al., 2012*), and

contribute themselves to the transcriptome (*Faulkner et al., 2009*; *Kelley and Rinn, 2012*; *Kapusta et al., 2013*). The influence of TEs on transcriptional programs has been studied in many broad contexts from the immune system (*Chuong et al., 2016*) to pregnancy (*Lynch et al., 2015*), to pluripotency (*Macfarlan et al., 2012*). Yet, ultimately, the regulatory impact of TEs has not been extensively studied at a genome-wide scale in a comparative framework.

Earlier work using a Down syndrome mouse model carrying a copy of human chromosome 21 showed that in general, genes on human chromosome 21 are similarly regulated regardless of whether the chromosome is in a human or a mouse cellular environment (*Wilson et al., 2008*). However, human-specific TEs become aberrantly activated (marked by H3K4me3) on human chromosome 21 in the mouse cellular environment (*Ward et al., 2013*). It was therefore speculated that host repressive factors needed to properly silence these human TEs may be absent in the mouse nuclear environment, which is ~60 million years diverged from human. This suggested a relatively rapid evolution of TE silencing mechanisms, in line with the evolutionary arms race hypothesis.

Indeed, due to the potential deleterious effects of new TE insertions, especially in the context of developmental regulatory programs, TE activity is thought to be tightly controlled by host repressive machinery. In particular, it has been shown that in embryonic stem cells (ESCs), KRAB-containing Zinc Finger genes (KRAB-ZNFs) can recognize TEs in a sequence-specific manner, recruit the co-factor TRIM28 (also known as KAP1), and histone methyltransferase SETDB1 (also known as ESET), to effect silencing through the deposition of the repressive histone three lysine nine trimethylation (H3K9me3) modification (*Matsui et al., 2010*; *Rowe et al., 2010*; *Najafabadi et al., 2015*; *Turelli et al., 2014*; *Friedli et al., 2014*; *Wolf and Goff, 2009*; *Schultz et al., 2002*). After the establishment of TE silencing by H3K9me3, TEs are subjected to de novo DNA methylation, which persists in somatic and germ cells (*Rowe and Trono, 2011*; *Quenneville et al., 2012*; *Rowe et al., 2013*; *Smith et al., 2014*; *Karimi et al., 2011*; *Walter et al., 2016*; *Bourc'his and Bestor, 2004*; *Barau et al., 2016*). Some TEs can evade detection by host KRAB-ZNFs by acquiring mutations. It was thus hypothesized that the host repressor mechanisms need to evolve rapidly in order to maintain genome integrity (*Kapopoulou et al., 2016*; *Huntley et al., 2006*; *Thomas and Schneider, 2011*).

TEs are silenced by KRAB-ZNFs particularly in early embryonic development (*Chuong et al., 2017*), their most likely niche for retrotransposition (*Richardson et al., 2017*); however, it is difficult to obtain biological material from humans and other primates to determine how this process evolves. Induced pluripotent stem cells (iPSCs) provide a model of the cells of the embryonic inner cell mass to sidestep this challenge. This cell type is of further interest as we know of examples of primate-specific TEs that are not silenced, but actively transcribed and functionally relevant in this cell type in human (notably HERVH elements) (*Klawitter et al., 2016*; *Santoni et al., 2012*; *Fort et al., 2014*; *Lu et al., 2014*; *Wang et al., 2014*; *Grow et al., 2015*; *Wissing et al., 2012*). This might imply specificity of TE regulatory mechanisms, and hints at the fact that evolutionarily recent TE sequence could play a role in differentiating gene regulatory networks between species.

The establishment of a panel of non-retroviral reprogrammed iPSCs from multiple humans and chimpanzees has allowed us to comprehensively characterize the extent of TE silencing in both species (*Gallego Romero et al., 2015*; *Banovich et al., 20162018*; *Burrows et al., 2016*). In order to gain insight into how TE regulation, specifically TE silencing, evolves between humans and chimpanzees, we globally profiled the predominant output of KRAB-ZNF-targeted silencing mechanisms in pluripotent cells: the histone modification H3K9me3. Our goals were to identify inter-species differences in TE silencing patterns, characterize gene regulatory divergence that can potentially be explained by differences in TE silencing between species, and develop hypotheses regarding the mechanisms that underlie such inter-species differences in TE regulation. Our observations led us to conclude that, at least in our comparative iPSC system, differences in TE silencing do not drive substantial gene regulatory divergence between these two closely related species.

## Results

To perform a comparative study of TE silencing, we characterized genome-wide distributions of the repressive histone modification H3K9me3 in iPSCs from 7 chimpanzees and 10 humans (*Figure 1A*, Table 1 in *Supplementary file 1*). The iPSC lines we used were deeply characterized in this study and previously (see Materials and methods; and [*Gallego Romero et al., 2015*; *Banovich et al.,*

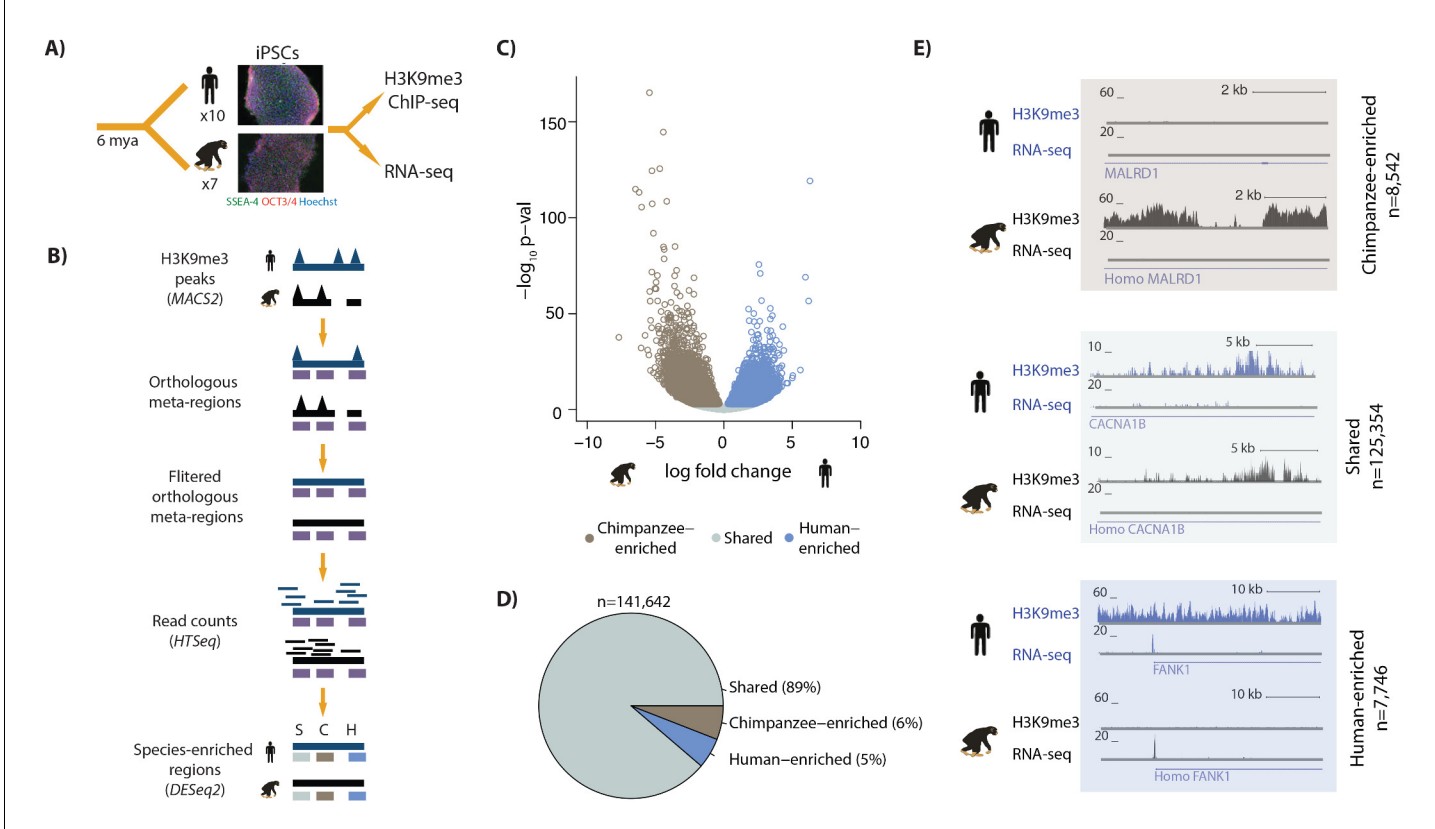

**Figure 1.** Majority of H3K9me3 regions are similarly enriched in human and chimpanzee iPSCs. (A) Experimental design of the study. (B) A heuristic description of the computational pipeline to identify regions of differential H3K9me3 enrichment between humans and chimpanzees. Human genome, ChIP-seq peaks and ChIP-seq reads (dark blue); chimpanzee genome, ChIP-seq peaks and ChIP-seq reads (black), orthologous H3K9me3 regions (purple). (C) Volcano plot representing the regions similarly enriched in both species (grey: Shared), enriched in human (light blue), or enriched in chimpanzee (brown) at 1% FDR. (D) Pie-chart representing the proportion of regions within the Shared, Human-enriched and Chimpanzee-enriched categories. (E) Examples of individual loci representing each of the three defined categories. The top track represents sequencing reads from H3K9me3 ChIP-seq experiments, and the bottom track gene expression levels of the *MALRD1*, *CACNA1B* and *FANK1* genes determined by RNA-seq. Also see *Figure 1—figure supplements 1–14*.

DOI: https://doi.org/10.7554/eLife.33084.002

The following figure supplements are available for figure 1:

**Figure supplement 1.** Newly generated human fibroblast-derived iPSCs have normal karyotypes and are able to differentiate into each of the three germ layers.

DOI: https://doi.org/10.7554/eLife.33084.003

**Figure supplement 2.** Human fibroblast-derived iPSCs exhibit gene expression profiles consistent with pluripotent cells.

DOI: https://doi.org/10.7554/eLife.33084.004

**Figure supplement 3.** Human and chimpanzee iPSCs express key pluripotency factors.

DOI: https://doi.org/10.7554/eLife.33084.005

**Figure supplement 4.** Number of ChIP and Input sequencing reads are similar across species.

DOI: https://doi.org/10.7554/eLife.33084.006

**Figure supplement 5.** H3K9me3 ChIP-seq reads are enriched at *ZNF* genes in both species.

DOI: https://doi.org/10.7554/eLife.33084.007

**Figure supplement 6.** The number of ChIP-seq peaks is similar in human and chimpanzee.

DOI: https://doi.org/10.7554/eLife.33084.008

**Figure supplement 7.** ChIP-seq peak numbers approach saturation with increasing read depth.

DOI: https://doi.org/10.7554/eLife.33084.009

**Figure supplement 8.** Reciprocal peak mapping retains ~90% of ChIP-seq peaks.

DOI: https://doi.org/10.7554/eLife.33084.010

**Figure supplement 9.** ChIP-seq data clusters by species.

DOI: https://doi.org/10.7554/eLife.33084.011

*Figure 1 continued on next page*

*Figure 1 continued*

**Figure supplement 10.** ChIP-seq data separates by species.

DOI: https://doi.org/10.7554/eLife.33084.012

**Figure supplement 11.** Inter-species variation is greater than intra-species variation.

DOI: https://doi.org/10.7554/eLife.33084.013

**Figure supplement 12.** Differential enrichment analysis identifies H3K9me3 regions enriched in each species.

DOI: https://doi.org/10.7554/eLife.33084.014

**Figure supplement 13.** Proportion of regions differentially enriched between species is robust with respect to the filtering approach used.

DOI: https://doi.org/10.7554/eLife.33084.015

**Figure supplement 14.** Majority of H3K9me3 regions differentially enriched between species have a small effect size.

DOI: https://doi.org/10.7554/eLife.33084.016

*20162018*; *Burrows et al., 2016*]). Briefly, all cell lines were confirmed to have a normal karyotype (*Figure 1—figure supplement 1A*), have the ability to form embryoid bodies that consist of cells from the three germ layers (*Figure 1—figure supplement 1B*), display global gene expression signatures consistent with pluripotent cells (*Figure 1—figure supplement 2*), and express key pluripotency factors detectable by immunocytochemistry (*Figure 1—figure supplement 3*). All samples were processed in species-balanced batches at each experimental step (Table 2 in *Supplementary file 1*).

We characterized the distribution of H3K9me3 in the 17 iPSC lines by performing chromatin immunoprecipitation followed by high-throughput sequencing (ChIP-seq). In order to increase genome-mapping confidence, paired-end sequencing libraries were prepared, and only high-quality, properly-paired reads retained (see Materials and methods and *Figure 1—figure supplement 4* and Table 3 in *Supplementary file 1*). As a first measure of quality of our data, consistent with previous reports, we observed an enrichment of H3K9me3 ChIP-seq reads at *ZNF* genes in both species (*Figure 1—figure supplement 5*)(*Roadmap Epigenomics Consortium et al., 2015*).

We identified regions of broad H3K9me3 ChIP enrichment in each individual independently (at an FDR of 10%; see Materials and methods and *Figure 1—figure supplement 6*). A read sub-sampling analysis indicates that we sequenced our samples to a depth that allows us to detect most enriched regions (*Figure 1—figure supplement 7*). In order to compare H3K9me3 enrichment across species, we focused exclusively on reciprocally best match orthologous regions in human and chimpanzee (see Materials and methods). We excluded from our data peaks that mapped outside those orthologous genomic regions (approximately 10% of the ChIP-seq data were excluded; *Figure 1—figure supplement 8* and Table 3 in *Supplementary file 1*). We further filtered the data based on overall genome mappability, to exclude regions in which sequencing reads can be mapped to one species at a substantially higher probability than the other (using a cutoff of 0.8 in both species; see Materials and methods). We defined orthologous H3K9me3 ChIP-seq regions as those where a ChIP-seq peak, contained within orthologous human-chimpanzee genomic regions, was identified in at least one individual, regardless of species (see Materials and methods). We refer to this combined set as 160,113 orthologous ChIP-seq regions (rather than ChIP-seq peaks), because these regions are defined across all individuals, while the ChIP-seq peak may have been identified in only a subset of individuals.

The definition of ChIP-seq regions allows a quantitative comparison of H3K9me3 enrichment across species. We therefore calculated the number of sequencing reads that were mapped to each orthologous ChIP-seq region in each individual. By using this approach, we sidestep the difficult challenge of accounting for incomplete power to detect 'significant' ChIP-seq peaks in each individual or species independently. To avoid considering regions where we clearly have insufficient power to compare read counts across species, we included only the 150,390 regions with at least one read count in more than half of the individuals, regardless of species.

We considered overall properties of the read count data in the ChIP-seq orthologous regions. As expected, data from different individuals cluster by species (with the exception of one human individual, which therefore was removed from subsequent analysis; *Figure 1—figure supplements 9–10*), and inter-species variation in read counts is somewhat greater than intra-species variation (Spearman's correlation in human vs. human = 0.90, chimpanzee vs. chimpanzee = 0.89, and human vs. chimpanzee = 0.80, *Figure 1—figure supplement 11*). We then used the framework of a linear

model with a fixed effect for species to identify orthologous ChIP-seq regions with differences in H3K9me3 enrichment between humans and chimpanzees (*Figure 1B*, *Supplementary file 2*, and Materials and methods). At FDR < 0.01, we classified 16,288 (11%) regions as differentially enriched between the two species (*Figure 1C–D*, *Figure 1—figure supplement 12* and see examples in *Figure 1E*). The proportion of differentially enriched regions is robust with respect to our data filtering choices (*Figure 1—figure supplement 13* and Table 4 in *Supplementary file 1*), and the choice of approach to analyze the data (Table 5 in *Supplementary file 1*).

## Orthologous TEs tend to be silenced more often than species-specific TEs

To compare TE silencing by H3K9me3 across species, we identified a comprehensive set of 4,036,865 autosomal orthologous TEs in the human and chimpanzee genomes (80.2% of all annotated human TEs, see Materials and methods and *Figure 2A*, *Figure 2—figure supplement 1*). We considered the ChIP-seq data in the context of the orthologous TEs, and observed that 12% of orthologous TEs (436,049 instances) overlap an orthologous H3K9me3 region with at least 50% of their length (denoted 'overlapping', *Figure 2B* and *Supplementary file 3*; we note that our subsequent conclusions are robust with respect to the specific cutoff used to classify a TE as overlapping with H3K9me3 regions). The TEs that do not overlap with an H3K9me3 region ('non-overlapping'; 3,600,816 instances), and are putatively not silenced by this repressive mechanism, are an equally interesting category of elements and are therefore included in subsequent analyses. Consistent with previous reports (*Turelli et al., 2014*; *Karimi et al., 2011*), we found enrichment of LTRs (Pearson's Chi-squared test; p<0.001), and SVA elements (p<0.001), among TEs that overlap H3K9me3 regions (*Figure 2C*).

Given the proposed evolutionary arms race between TEs and the host repressive machinery (*Jacobs et al., 2014*), we sought to investigate the within-species patterns of silencing at both orthologous and non-orthologous TEs. We further categorized non-orthologous TEs into those that are only annotated in that species (defined as 'species-specific TEs'; for example, SVA-E and SVA-F, which are only annotated in human; Table 6 in *Supplementary file 1*). Generally, we found that the same TE classes that most frequently overlap with H3K9me3 regions in the orthologous TE set, also most frequently overlap with H3K9me3 in the non-orthologous TE set (*Figure 3*). However, within each species, the SVA and LINE TE classes show a reduction in the proportion of TEs overlapping H3K9me3 regions in the non-orthologous, and species-specific categories (p (SVA) <0.001; p (LINE) <0.001); *Figure 3—figure supplement 1*). The magnitude of the effect differs between these two TE classes: For example, 85% of orthologous SVA elements overlap H3K9me3, but only 41% of non-orthologous SVA elements overlap the regions marked by H3K9me3. In contrast, 17% and 14% orthologous and non-orthologous LINE elements overlap H3K9me3 regions, respectively. In this case, the difference in effect size could reflect heterogeneity in silencing amongst the diverse elements within the large LINE class. We also found a reduction of TE and H3K9me3 overlap in human LTRs (p<0.001), but an increase in H3K9me3 overlap with LTRs that are chimpanzee-specific (p<0.001; *Figure 3*). In general, these observations are consistent with the notion that more recent TEs are generally less likely to be silenced, in line with the evolutionary arms race theory (*Kapopoulou et al., 2016*; *Huntley et al., 2006*; *Thomas and Schneider, 2011*).

## The majority of orthologous TEs are similarly silenced in human and chimpanzee

To determine the extent to which orthologous TEs are similarly silenced in both species, we performed a comparative quantitative analysis of the ChIP-seq read counts in the orthologous TEs that overlap H3K9me3 regions. As mentioned previously, our comparative analysis is not dependent on the identification of 'significant' ChIP-seq peaks in both species. Instead, we tested for evidence of differences in H3K9me3 read counts between humans and chimpanzees in all ChIP-seq regions. When we could not reject the null hypothesis (at FDR < 0.01) that similar numbers of H3K9me3 ChIP-seq reads are mapped to orthologous TEs, we referred to the silencing status of these TEs as 'shared'. We thus classified TEs into the following categories (*Supplementary file 3*) : (i) TEs that do not overlap with an orthologous H3K9me3 region (non-overlapping); (ii) TEs that are associated with

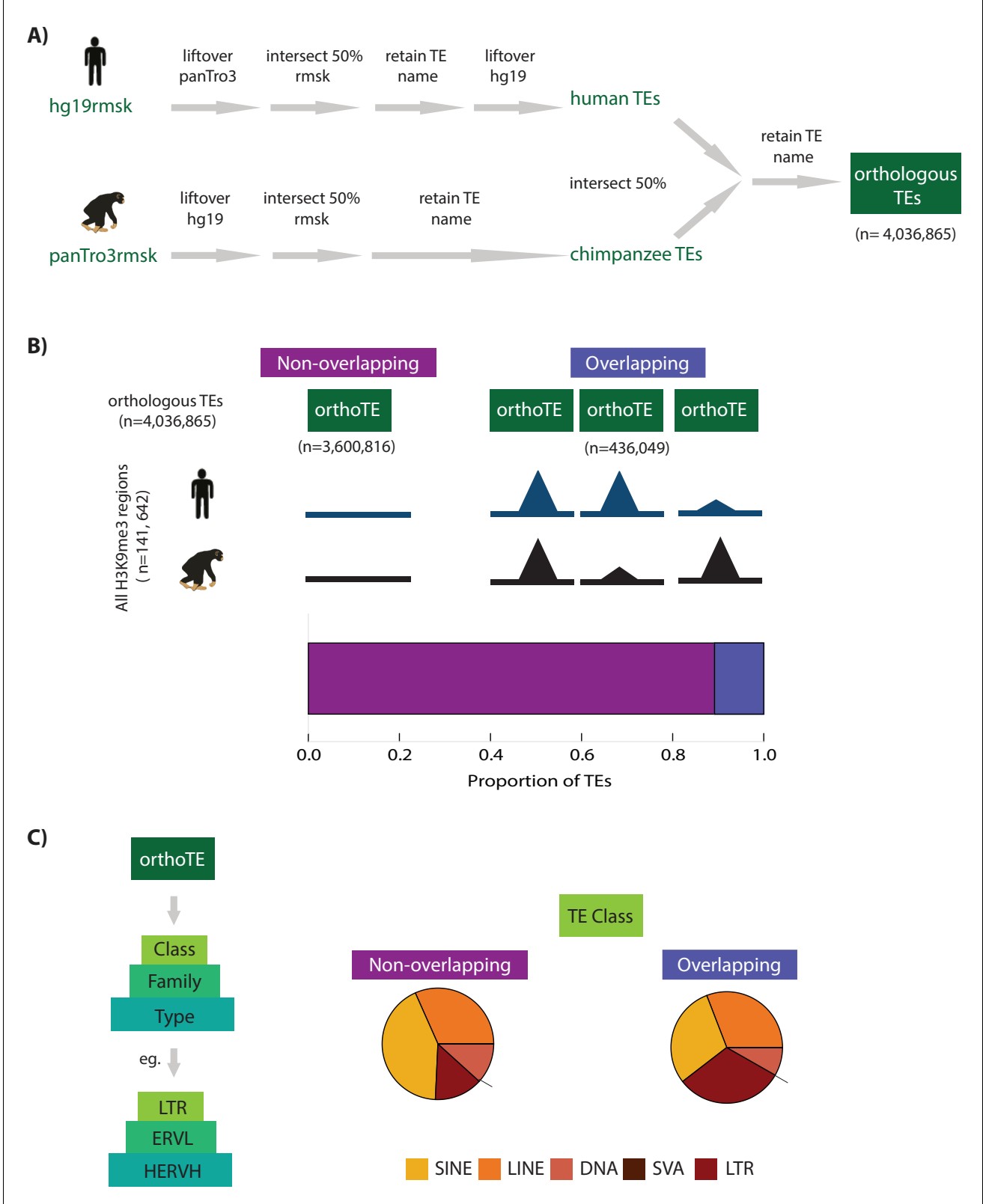

**Figure 2.** Transposable element silencing is associated with particular transposable element classes. (**A**) Bioinformatic pipeline used to identify TEs that are orthologous between humans and chimpanzees using RepeatMasker (rmsk) tracks in each species. (**B**) The proportion of all TEs that overlap an orthologous H3K9me3 region (H3K9me3 peak identified in at least one individual of one species) (purple) compared to those that do not overlap an orthologous H3K9me3 region (magenta). (**C**) Each orthologous TE instance is further hierarchically classified into Class, Family and Type. The proportion

*Figure 2 continued on next page*

*Figure 2 continued*

of all orthologous TEs belonging to the five main TE classes (SINE: yellow, LINE: orange, DNA: peach, SVA: brown, LTR: red) that do not overlap an H3K9me3 region in either species, and the proportion of TEs that do overlap a H3K9me3 region. The small number of SVA elements means that this class is not visible in the pie chart. Also see *Figure 2—figure supplement 1*.

DOI: https://doi.org/10.7554/eLife.33084.017

The following figure supplement is available for figure 2:

**Figure supplement 1.** Orthologous TEs reflect the distribution of TE classes in the human and chimpanzee genome.

DOI: https://doi.org/10.7554/eLife.33084.018

similar enrichment of H3K9me3 in both species (shared); (iii) TEs that are associated with significantly higher enrichment of H3K9me3 in humans or (iv) in chimpanzees.

We observed that the proportion of TEs that overlap with H3K9me3 regions differs substantially across different TE classes (for example, 60% of SVA elements overlap H3K9me3 regions compared to only 10% of DNA transposons). Yet, the proportion of TEs for which we have evidence for inter-species differences in silencing is consistent, regardless of TE class (*Figure 4A*). Indeed, while ultimately most TEs (88%) do not overlap H3K9me3 regions, when TEs do overlap H3K9me3 regions, they are generally (82% of TEs) silenced to the same extent in the two species. Our analysis includes hundreds of thousands of TEs that overlap H3K9me3 regions; thus even just 18% of TEs that overlap H3K9me3 and are silenced differently across species still comprise of tens of thousands of instances (53,201 TEs enriched for H3K9me3 in human, and 28,110 TEs in chimpanzee).

Thousands of instances of differential silencing notwithstanding, the overall emerging pattern is that most TEs are silenced similarly between species. Yet, it is challenging to draw conclusions based on the inability to reject a null hypothesis of no differences between the species. We therefore used an alternative approach, based on an assumption of a unimodal distribution of effects (ashr) (*Stephens, 2017*) to obtain a more sensitive estimate of the proportion of inter-species differences at different effect sizes. This unimodal assumption in an Empirical Bayes framework produces smaller (more conservative) estimates of the proportion of tests under the null through adaptive shrinkage. This test is therefore more sensitive to smaller differences in effect size across species. Using this approach, we again find a high degree of conservation in TE silencing, with practically all inter-species differences in silencing associated with small effect sizes (smaller than two fold; *Figure 1—figure supplement 14*).

Given the diversity of TEs within a class, we proceeded by considering a higher resolution classification of TEs to the 11 most predominant TE families representing each of the 5 TE classes, and then to TE types within specific families (*Figure 2*). There is some degree of heterogeneity in inter-species silencing differences ('effect size') between different TE families (75–84% depending on the family; *Figure 4—figure supplement 1*). We particularly focused on the LTR and LINE classes, given the previously reported association of these TEs with TRIM28-mediated silencing mechanisms (*Turelli et al., 2014*; *Castro-Diaz et al., 2014*). Human LTRs are comprised of HERV elements which can be alternatively classified into Class I (HERV-F,H,I,E,R,W families), Class II (HERV-K), and Class III (HERV-L), based on the original retrovirus that integrated into the genome (*van der Kuyl, 2012*). Using a subset of these HERV types curated by Turelli *et al.*, where the annotation is likely to be correct, we observed considerable heterogeneity in the frequency with which different HERV types overlap H3K9me3 regions (*Figure 4—figure supplement 2A*). Notably, less than 20% of 2,079 orthologous HERVH elements, which are generally known to be expressed in pluripotent cells, overlap with H3K9me3 regions (compared to 80% of 3,484 THE1B elements, which are not known to be expressed in pluripotent cells, for example). This heterogeneity is also reflected in the proportion of TEs that are similarly silenced in the two species (55–95% sharing, *Figure 4—figure supplement 2B*). However, the total number of elements within the more divergent TE types is small (fewer than 100 TE instances per category with less than 75% sharing). In contrast, when turning our attention to a subset of mammalian-specific L1 LINE elements (*Castro-Diaz et al., 2014*), we found less variability in overlap with H3K9me3 across types (5–25% are overlapping; *Figure 4—figure supplement 3A*). However, of those L1 elements that overlap H3K9me3, the majority (82% of 21,710 elements) are similarly silenced in both species (*Figure 4—figure supplement 3B*). These results suggest that there is some degree of sequence specificity in the silencing mechanisms across TE types.

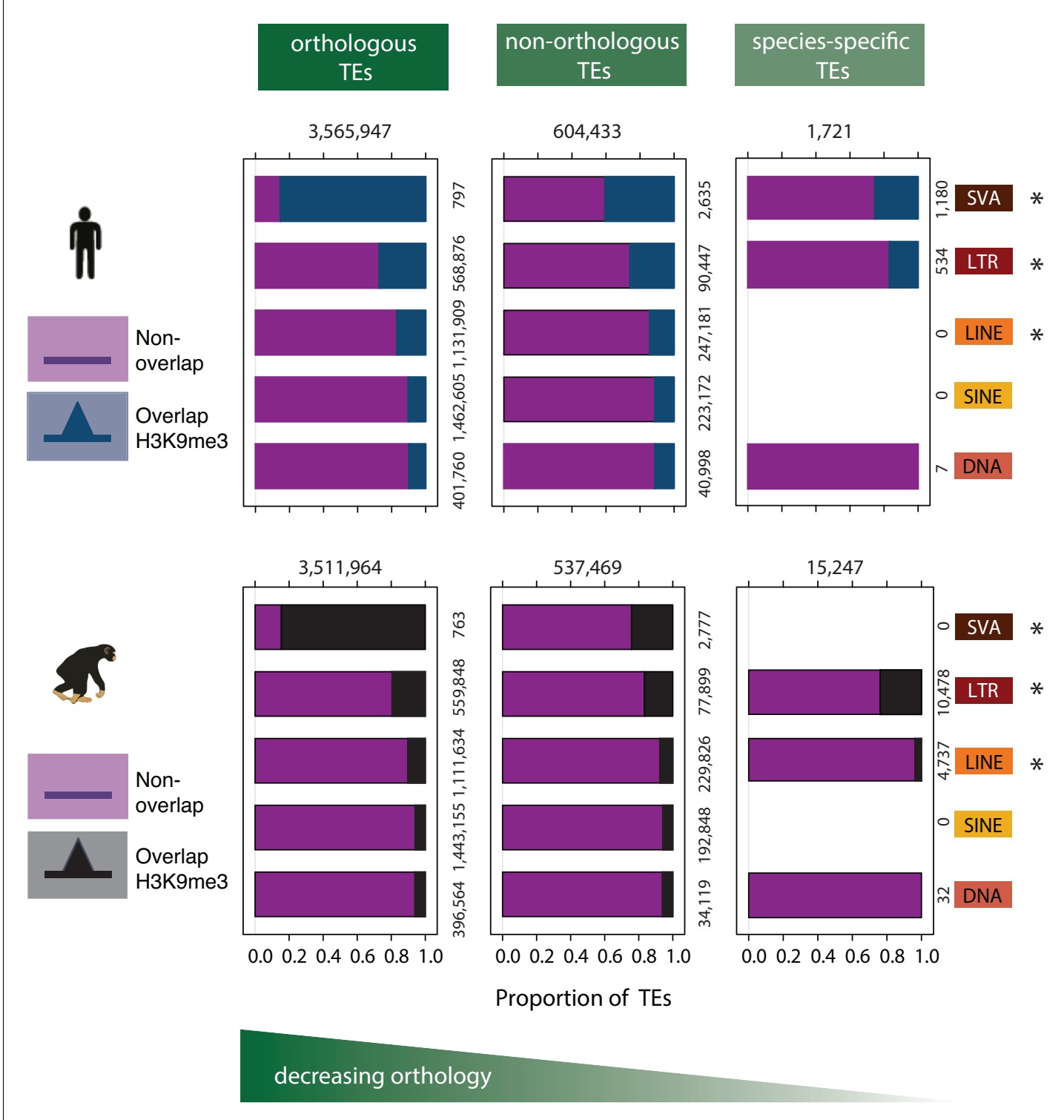

**Figure 3.** Patterns of preferential TE class-based silencing are maintained as orthology decreases. Within-species analysis of the proportion of orthologous TEs (left panel), TEs that are not orthologous between the two species (middle panel), and TEs that are not orthologous and whose type is annotated in a single species (right panel), that overlap any H3K9me3-enriched region identified in that species. Magenta represents those TEs that do not overlap with any H3K9me3 region identified in that species, blue represents TEs that overlap any H3K9me3 regions in human, and black those TEs that overlap with H3K9me3 regions in chimpanzee. The absence of data in the species-specific category reflects that there are no species-specific annotations in these TE classes. Asterisk denotes when there is a significant difference between orthology categories (Pearson's Chi-squared test; p<0.001). Also see *Figure 3—figure supplement 1*.

*Figure 3 continued on next page*

*Figure 3 continued*

DOI: https://doi.org/10.7554/eLife.33084.019

The following figure supplement is available for figure 3:

**Figure supplement 1.** Species-specific SVA elements overlap H3K9me3 regions less frequently than orthologous SVAs.

DOI: https://doi.org/10.7554/eLife.33084.020

## Silenced TEs tend to have similar functional properties across species

We next sought to identify properties that might distinguish TEs based on our silencing classifications (shared, species-specific, or not silenced). We first considered the length of a TE, which may relate to its transposition competency and hence need for silencing. On average, TEs silenced in at least one species are longer than those that do not overlap orthologous H3K9me3 regions (Wilcoxon-rank sum test; p<0.001; *Figure 5A*). To confirm that longer TEs do not overlap H3K9me3 more often by chance alone, we artificially increased the length of the non-overlapping TEs to match the median length of TEs marked by H3K9me3 in both species. The artificially longer TEs do not overlap with H3K9me3 regions more often, suggesting that TE elements are silenced in a non-random manner, which cannot be simply explained by length in itself. We found the same pattern when we analyzed TEs by class, which in itself is associated with different lengths (p<0.001; *Figure 5—figure supplement 1A*). Interestingly, across TE families there is a strong correlation between the proportion of TEs that overlap H3K9me3 regions and the median length of the TEs in the family (Pearson's correlation = 0.93; *Figure 5—figure supplement 1B–C*). However, within families the median length of TEs that do not overlap H3K9me3 is highly similar to that of TEs that do (Pearson correlation for median TE lengths across families = 0.99; *Figure 5—figure supplement 1D*), again suggesting that this is not a feature of length itself. When we stratified TEs by the magnitude of inter-species silencing divergence (absolute effect size), we found that shorter TEs are associated with higher silencing divergence (p<0.001; *Figure 5B*). Put together, these observations suggest that longer TEs may be silenced more often because they are more likely to have unwanted regulatory potential, and are therefore more likely to contain sequences that recruit silencing machinery. This is consistent with the finding that more KRAB-ZNF proteins (sequence-specific mediators of TE silencing) bind longer TEs (*Imbeault et al., 2017*).

To gain insight into the relationship between TE silencing and TE transposition and mobility, we asked whether TE copy number is correlated with silencing. Across TE families, we found only a weak correlation between TE copy number and the proportion that overlap with H3K9me3 regions (Pearson's correlation = −0.47; permutation p=0.15; *Figure 5—figure supplement 2*). Moreover, in contrast to the general trend, TEs in the two smallest families (780 SVAs and 7,983 ERVKs) overlap H3K9me3 regions most often (*Figure 5—figure supplement 2*).

Given that TEs can acquire mutations and transpose over time, we compared the sequence divergence between each individual TE instance and the corresponding consensus TE sequence (see Materials and methods). We found that TEs that are silenced similarly in both species are evolutionarily less diverged from the consensus sequence than TEs that do not overlap H3K9me3 regions (p<0.001; *Figure 5C*). This pattern is observed within the LTR, LINE, SINE and DNA TE classes (p<0.001; *Figure 5—figure supplement 3A*), but this pattern does not hold within all TE families, as might be expected considering the diversity of TE types and families within a class (*Figure 5—figure supplement 3B*). Across all TEs, we did not find a difference in sequence divergence between TEs that are silenced in just one or both species (*Figure 5C*) but stratification of TEs by the magnitude of the inter-species silencing effect reveals that when inter-species differences in silencing are larger, the sequences of the TEs are more diverged from the consensus sequence (p<0.001; *Figure 5D*).

We next considered proximity of TEs to genes by determining the distance between each TE and the closest orthologous annotated Transcription Start Site (TSS). TEs that are silenced in both species are significantly further away from the TSS than TEs that are preferentially silenced in one species, or those that do not overlap with H3K9me3 regions (p<0.001; *Figure 5E*). Again, this pattern is observed within the LTR, LINE, SINE and DNA classes, but not within all TE families (p<0.001; *Figure 5—figure supplement 4*). TEs that show greater divergence in silencing across species are also more likely to be closer to the TSS than TEs with lower divergence (p<0.001; *Figure 5F*). These

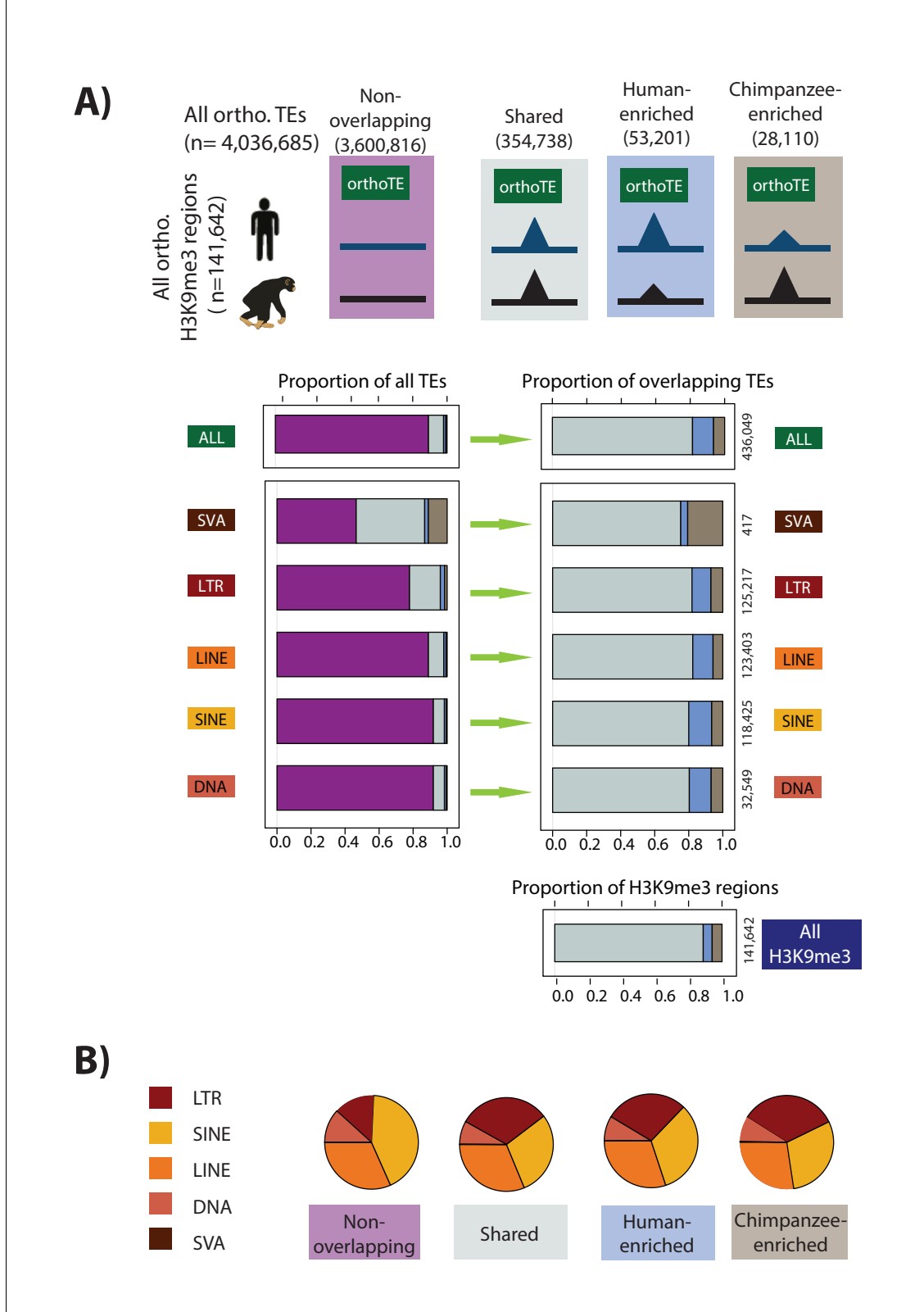

**Figure 4.** Majority of orthologous TEs may be similarly silenced in humans and chimpanzees. (**A**) The proportion of orthologous TEs in each class that do not overlap an orthologous H3K9me3 region (magenta), show similar H3K9me3 enrichment in both species (grey: Shared), are enriched in human (blue), or are enriched in chimpanzee (brown) are shown. Of those orthologous TEs that overlap an orthologous H3K9me3 region, the distribution between those that are Shared, Human-enriched, or Chimpanzee-enriched is shown in the right panel. (**B**) The proportion of orthologous TE elements

*Figure 4 continued on next page*

*Figure 4 continued*

belonging to each class within each of the four H3K9me3 silencing categories (LINE: orange, SINE: yellow, LTR: red, SVA: brown, DNA: peach). The small number of SVA elements means that this class is not visible in the pie chart. Also see *Figure 4—figure supplements 1–3*.

DOI: https://doi.org/10.7554/eLife.33084.021

The following figure supplements are available for figure 4:

**Figure supplement 1.** TEs across families have similar levels of H3K9me3 enrichment between species.

DOI: https://doi.org/10.7554/eLife.33084.022

**Figure supplement 2.** HERV TE types have variable overlap with orthologous H3K9me3 regions and variable inter-species enrichment.

DOI: https://doi.org/10.7554/eLife.33084.023

**Figure supplement 3.** LINE-1 TE types are similarly enriched for H3K9me3 in humans and chimpanzees.

DOI: https://doi.org/10.7554/eLife.33084.024

results suggest that, across families, there are some differences in the functional properties of silenced TEs.

In order to determine whether silenced TEs are also associated with other genomic features, we overlaid orthologous TEs with chromatin state annotations defined in human ESCs (*Ernst and Kellis, 2010*). The majority of TEs, regardless of silencing status, are associated with chromatin regions annotated as 'heterochromatin and low signal'. We found that TEs that are preferentially silenced in human overlap annotated regulatory regions less frequently compared with TEs that are preferentially silenced in chimpanzee (p<0.001; *Figure 5G* and *Figure 5—figure supplement 5*). We also considered published ChIP-seq and DNase I hypersensitivity (DHS) data in H1 hESCs. We found that a relatively small fraction of TEs overlap regions of open chromatin or ChIP-seq peaks: Less than 20% of TEs overlap active histone marks and less than 1% of TEs overlap DHS. We found that TEs that are preferentially silenced in human overlap regions of active gene regulation less often than TEs preferentially silenced in chimpanzee. The active chromatin annotations we considered include marking by H3K27ac, DHS, enhancer elements (marked by H3K4me1, p300) and regions marked by RNA polymerase II occupancy (for all comparisons p<0.01; *Figure 5—figure supplement 6*). Notably, there is no significant difference between TEs preferentially silenced in either species with respect to overlap with the active promoter-associated mark, H3K4me3.

## TE silencing is not a major driver of inter-species gene expression divergence

To determine the potential impact of TE silencing on the regulation of neighboring genes, we characterized gene expression levels in the same iPSC lines from both species (using RNA-sequencing; see Materials and methods, *Figure 6A*, *Figure 6—figure supplement 1A* and Tables 2-3 in *Supplementary file 1*). After restricting the data to a set of human-chimpanzee orthologous exons (*Blekhman et al., 2010*) the RNA-seq data clusters primarily by species, and then by population (i.e. Caucasian or Yoruba/cell type of origin; *Figure 6—figure supplement 1B*). Indeed, after filtering lowly expressed genes, species emerges as the primary driver of variation in the RNA-seq data, as expected (*Figure 6—figure supplement 1C*).

To characterize the level of TE silencing associated with each orthologous gene, we defined a *cis*-regulatory region 10 kb upstream of the TSS of each gene that includes at least one annotated TE (Materials and methods and *Figure 6A*). As expected, we observed that gene expression levels decrease when TEs in the *cis*-regulatory window overlap with H3K9me3 regions (p<0.01; *Figure 6B*). This observation is consistent with the general reported role of the H3K9me3 histone modification in gene regulation (*Bilodeau et al., 2009*; *Mozzetta et al., 2015*). The effect of TE silencing on gene expression is recapitulated when extending the window size to 20 kb, while TEs silenced within 1 kb upstream of the TSS have a stronger effect on gene expression (*Figure 6—figure supplement 2*). We then asked whether inter-species differences in TE silencing are associated with expression divergence of nearby genes.

To obtain a quantitative measure of overall TE silencing in each species, we summed H3K9me3 ChIP-seq read counts from orthologous H3K9me3 regions overlapping TEs within 10 kb of annotated genes. We used the same linear model framework described above to estimate inter-species difference in H3K9me3 counts, within a 10 kb window upstream of the TSS. Of the windows that contain at least one orthologous H3K9me3 region overlapping a TE, 79% have similar levels of

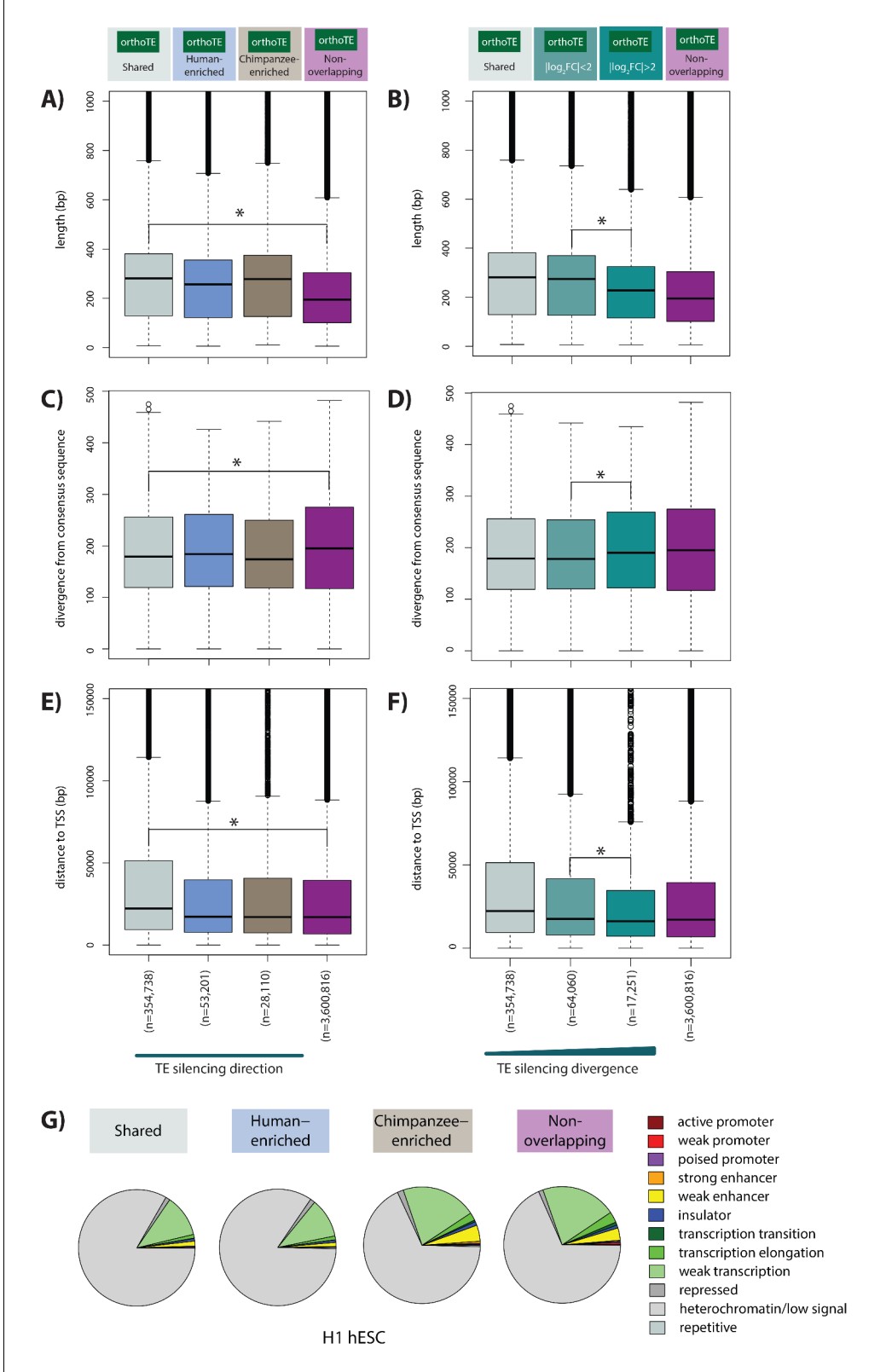

**Figure 5.** Majority of silenced orthologous TEs have similar properties in humans and chimpanzees. (**A**) The length of all orthologous TE instances occurring within Shared, Human-enriched and Chimpanzee-enriched H3K9me3 silencing categories. (**B**) The length of all orthologous TEs stratified by inter-species divergence in effect size. (**C**) The sequence divergence from the consensus TE sequence of TEs categorized as in (**A**). (**D**) The sequence divergence of TEs categorized as in (**B**). (**E**) The distance between each TE and the closest orthologous TSS categorized as in (**A**). (**F**) The distance

*Figure 5 continued on next page*

*Figure 5 continued*

between each TE and the closest orthologous TSS categorized as in (**B**). Grey: Shared, blue: Human-enriched, brown: Chimpanzee-enriched, magenta: does not overlap an orthologous H3K9me3 region, light teal: log$_2$ fold change less than two between species, dark teal: log$_2$ fold change greater than two between species. (**G**) Orthologous TEs categorized by silencing status that overlap chromatin states annotated in the H1 hESC line. Asterisk denotes when there is a significant difference between the Shared and Non-overlapping, and high and low divergence categories (Wilcoxon-rank sum test; p<0.001). Also see *Figure 5—figure supplements 1–6*.

DOI: https://doi.org/10.7554/eLife.33084.025

The following figure supplements are available for figure 5:

**Figure supplement 1.** Longer TE families overlap with orthologous H3K9me3 regions more often than shorter families.

DOI: https://doi.org/10.7554/eLife.33084.026

**Figure supplement 2.** TE copy number is weakly anti-correlated with silencing by H3K9me3.

DOI: https://doi.org/10.7554/eLife.33084.027

**Figure supplement 3.** TEs, within a class, that are similarly enriched for H3K9me3 in both species tend to be less diverged from the consensus TE sequence than TEs that do not overlap H3K9me3 .

DOI: https://doi.org/10.7554/eLife.33084.028

**Figure supplement 4.** TEs, within a class, that are similarly enriched for H3K9me3 in both species tend to be further away from the nearest TSS than TEs that do not overlap H3K9me3 .

DOI: https://doi.org/10.7554/eLife.33084.029

**Figure supplement 5.** TEs preferentially silenced by H3K9me3 in human, are depleted for regions of active chromatin in human, relative to TEs preferentially silenced in chimpanzee.

DOI: https://doi.org/10.7554/eLife.33084.030

**Figure supplement 6.** TEs preferentially silenced by H3K9me3 in human are depleted for regulatory regions in human relative to TEs preferentially silenced in chimpanzee.

DOI: https://doi.org/10.7554/eLife.33084.031

H3K9me3 enrichment in human and chimpanzee (3,523), with 11% of regions enriched in human (499), and 10% of regions enriched in chimpanzee (463), at FDR of 1% (*Figure 6C*). We then considered inter-species differences in expression levels for genes associated with similar or species-specific H3K9me3 enrichment at TEs.

We found no significant difference in the expression divergence effect size between genes associated with species-specific or shared TE silencing, using 10, 20 or 40 kb windows upstream of the TSS (*Figure 6C* and *Figure 6—figure supplement 3*). We did find a weak, negative association between gene expression and silencing divergence when we considered silenced TEs that are located within 1 kb upstream of the TSS (only 165 genes in that set; *Figure 6—figure supplements 3* and *4*; Table 7 in *Supplementary file 1*). It is tempting to conclude that TEs (or silenced TEs) near the TSS can have a marked contribution to expression divergence; however, H3K9me3 enrichment in regions that overlap a 1 kb region around the TSS is similarly associated with expression divergence, irrespective of the presence of a TE (*Figure 6—figure supplement 4*). Our observations are robust with respect to a wide range of FDR cutoffs used to classify inter-species differences in silencing, suggesting that incomplete power is not a likely explanation for the overall lack of association between expression divergence and TE silencing (Table 8 in *Supplementary file 1*).

To further investigate the overall lack of association between silencing and gene expression divergence, we performed an analogous analysis, this time using gene expression divergence as the anchor. We found no association between silencing and inter-species expression divergence: genes that are classified as differentially expressed between humans and chimpanzees are not more likely to be associated with inter-species differences in H3K9me3 at TEs in their putative regulatory regions (*Figure 6D* and *Figure 6—figure supplement 5A*). Again, we could only find a relationship between silencing and expression divergence when considering H3K9me3 overlapping TEs within 1 kb upstream of the TSS. The observation of no association of inter-species expression divergence and TE silencing is robust with respect to a wide range of statistical cutoffs used to classify either differential expression or differential silencing (*Figure 6—figure supplement 5B* and Table 9 in *Supplementary file 1*). Indeed, when we considered the entire distribution, the correlation between inter-species effect sizes for gene expression divergence and H3K9me3 enrichment is weak (Pearson's correlation = −0.0005 in a 10 kb window; the correlation in divergence increases somewhat to −0.14 when considering a 1 kb window upstream of the TSS; *Figure 6—figure supplement 3*).

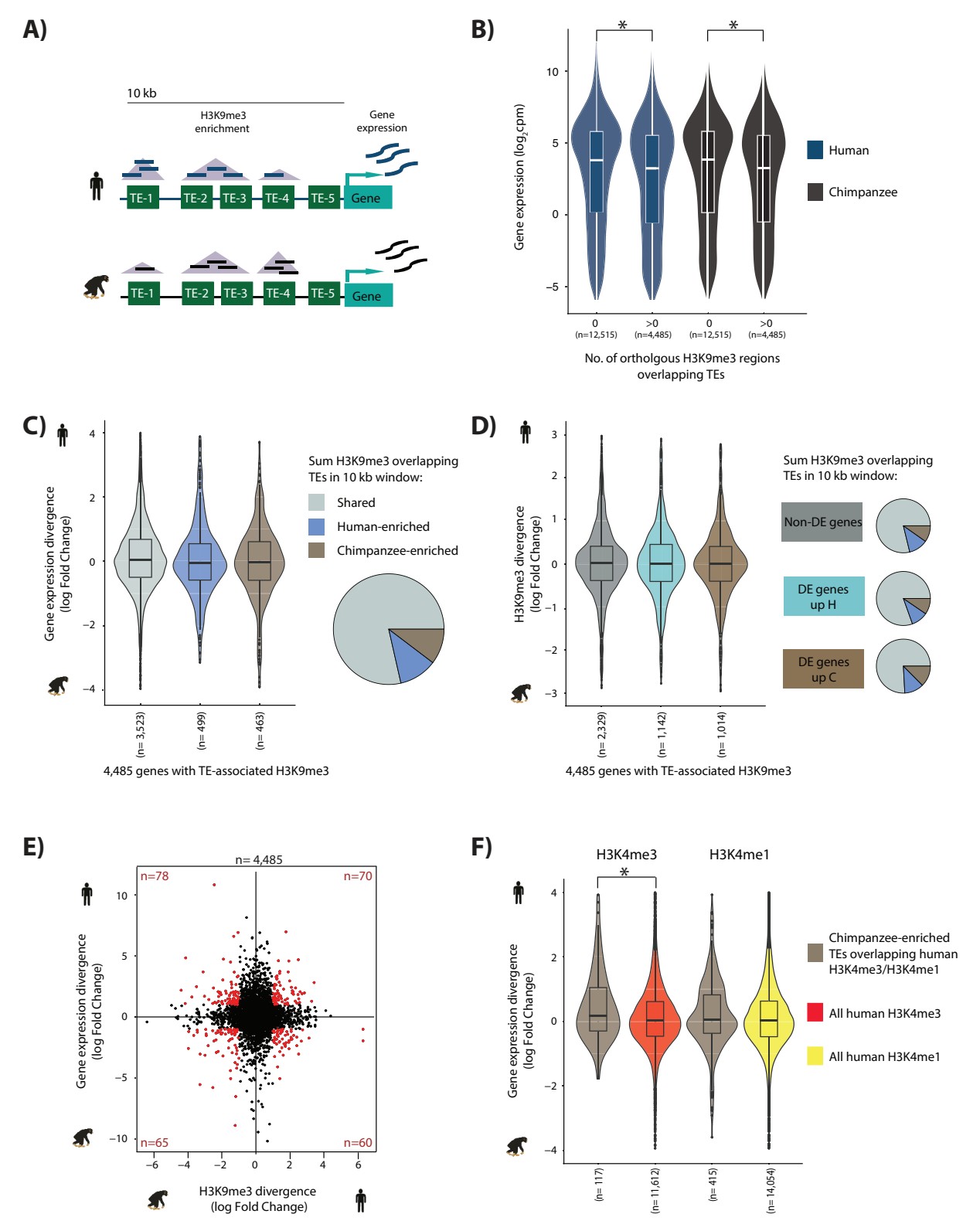

**Figure 6.** TE silencing divergence between species is not correlated with gene expression divergence. (**A**) Schematic representation of the experiment to determine how TE silencing affects nearby gene expression; orthologous TEs (green rectangles), orthologous H3K9me3 regions (purple triangles), human read counts (dark blue), and chimpanzee read counts (black). (**B**) Human and chimpanzee gene expression levels of orthologous genes with at least one TE present in a 10 kb window upstream of the TSS. Genes are categorized by the presence or absence of H3K9me3 regions overlapping TEs. *Figure 6 continued on next page*

*Figure 6 continued*

(C) Gene expression divergence between human and chimpanzee at genes where TE silencing in the upstream 10 kb region is classified as Shared, Human-enriched or Chimpanzee-enriched. The proportion of genes associated with each TE silencing category is represented by the pie chart. (D) Stratification of genes based on differential expression status: genes that are similarly expressed in human and chimpanzee (dark grey), those that are up-regulated in human (turquoise), and those that are up-regulated in chimpanzee (dark brown) (FDR < 1%). The proportion of TEs marked by H3K9me3 in a 10 kb window upstream of the TSS in each gene category that are Shared, enriched in human, and enriched in chimpanzee are represented as pie charts. Box plots represent human-chimpanzee divergence of H3K9me3 enrichment overlapping TEs for each gene category. (E) Divergence in H3K9me3-mediated TE silencing between species versus neighboring gene expression divergence. Genes which have a H3K9me3 log fold divergence of >1 or <-1, and a gene expression divergence log fold change of >1 or <-1 are shown in red. (F) Inter-species gene expression divergence of orthologous genes closest to TEs overlapping H3K9me3 regions enriched in chimpanzee, and overlapping human H3K4me3/H3K4me1 regions identified in H1 hESCs. The closest orthologous genes to all human H3K4me3/H3K4me1 regions in H1 cells are used for comparison. Asterisk denotes a significant difference between categories (Wilcoxon-rank sum test; p<0.01). Also see *Figure 6—figure supplements 1–10*.
DOI: https://doi.org/10.7554/eLife.33084.032

The following figure supplements are available for figure 6:

**Figure supplement 1.** RNA-seq data clusters by species.
DOI: https://doi.org/10.7554/eLife.33084.033

**Figure supplement 2.** TEs overlapping H3K9me3 regions upstream of genes associate with decreased gene expression.
DOI: https://doi.org/10.7554/eLife.33084.034

**Figure supplement 3.** TEs preferentially enriched for H3K9me3 in one species only have a minimal effect on gene expression divergence at the TSS.
DOI: https://doi.org/10.7554/eLife.33084.035

**Figure supplement 4.** H3K9me3 preferentially enriched in one species at the TSS associates with gene expression divergence.
DOI: https://doi.org/10.7554/eLife.33084.036

**Figure supplement 5.** Genes that are differentially expressed between human and chimpanzee are not associated with inter-species differences in H3K9me3 enrichment at TEs distal to the TSS.
DOI: https://doi.org/10.7554/eLife.33084.037

**Figure supplement 6.** TEs in human enhancer regions do not associate with inter-species gene expression divergence.
DOI: https://doi.org/10.7554/eLife.33084.038

**Figure supplement 7.** Non-orthologous TE silencing distal to the TSS does not consistently influence gene expression divergence between species.
DOI: https://doi.org/10.7554/eLife.33084.039

**Figure supplement 8.** Orthologous KRAB-ZNF genes overlap with orthologous H3K9me3 regions.
DOI: https://doi.org/10.7554/eLife.33084.040

**Figure supplement 9.** KRAB-ZNF genes are no more likely to be differentially expressed between species than all genes.
DOI: https://doi.org/10.7554/eLife.33084.041

**Figure supplement 10.** TEs preferentially silenced by H3K9me3 in chimpanzee are not more often associated with increased levels of H3K27me3 compared to TEs preferentially silenced by H3K9me3 in human.
DOI: https://doi.org/10.7554/eLife.33084.042

Moreover, we observed nearly as many genes whose expression divergence is consistent with the observed patterns of TE silencing divergence (namely, TE silencing associated with lower expression level) as those that show the opposite, inconsistent pattern (*Figure 6E*).

We looked at the relationship between expression divergence and TE silencing more closely by considering orthologous TEs that are most likely to regulate neighboring gene expression. Specifically, we focused on the 532 genes closest to TEs that overlap either enhancer (marked by H3K4me1) or promoter (marked by H3K4me3) regions in human, and yet are silenced in chimpanzee (*Figure 5—figure supplement 6*). The results are consistent with our observations for the entire set. We found a relationship between expression divergence and silencing of TEs that overlap the promoter-associated mark but not of TEs that overlap enhancers (typically further from the TSS; *Figure 6F* and *Figure 6—figure supplement 6*).

We next reasoned that perhaps TEs that are actively expressed in this cell type, in at least one species, are more likely to affect gene expression divergence. Indeed, TEs can be transcribed and contribute to lncRNA and neighboring gene expression (*Ramsay et al., 2017*; *Karimi et al., 2011*). Using our RNA-seq data, we identified 11 poly-adenylated TE types that are expressed in chimpanzee, and 7 that are expressed in human (Materials and methods; Table 10A in *Supplementary file 1*; note that this approach does not allow us to capture non-polyadenylated transcripts such as eRNAs). In both species, we identified the expression of HERVH elements, which are known to be expressed in human pluripotent cells. Of the nine TE types that are annotated in both species and expressed in

at least one species, 1,025 instances are contained within our orthologous TE set (Table 10B in *Supplementary file 1* and 123 instances overlap an H3K9me3 region. Only 14 of these TE instances are enriched for H3K9me3 in human, and 22 in chimpanzee (FDR < 0.01). These TEs are proximal (within 10 kb) of only 19 orthologous genes. Thus, due to the small numbers we are unable to properly test the association between TE silencing/expression and gene expression divergence. In any case, even if there is such an association in these cases, the small number of instances suggests that gene regulation by an actively expressed poly-adenylated TE in one species is not a major mechanism of gene expression divergence.

Further attempting to find a relationship between TE silencing and expression divergence, we hypothesized that perhaps imperfect annotation of orthologous TEs may affect our observations. A clear association between silencing of non-orthologous TEs and gene expression divergence would provide some measure of support for this possible explanation. We thus performed a similar analysis focused only on the non-orthologous TEs. In this analysis, we considered non-orthologous TEs that are located 10 kb upstream of annotated orthologous genes. We classified the non-orthologous TEs as 'silenced' or 'not silenced' based on all H3K9me3 regions identified in that species (we did not require H3K9me3 regions to be orthologous in this analysis). We found, once again, that there is no association between TE silencing and expression divergence (*Figure 6—figure supplement 7*). A significant effect is only consistently observed when considering a 1 kb window immediately upstream of the TSS (a set that includes 289 genes; *Figure 6—figure supplement 7*). In other words, TE silencing by H3K9me3, regardless of orthology, has a modest effect on gene expression divergence in human and chimpanzee iPSC lines.

We next focused our attention on known mediators of TE silencing, the KRAB-ZNF genes, in an attempt to identify species-specific regulatory effects of TE silencing. KRAB-ZNF genes, in addition to directing the repression of TEs, are themselves marked by H3K9me3 (*Roadmap Epigenomics Consortium et al., 2015*; *O'Geen et al., 2007*). We intersected a curated list of 256 KRAB-ZNF genes (*Kapopoulou et al., 2016*), filtered for orthology, with our list of orthologous H3K9me3 regions. We found that 104 (41%) of these genes overlap with orthologous H3K9me3 regions (*Figure 6—figure supplement 8A*). The proportion of KRAB-ZNFs with similar H3K9me3 enrichment in both species (78% of overlapping genes or 81 instances) is comparable to that observed for TEs (*Figure 6—figure supplement 8B*). 17 KRAB-ZNFs are enriched for H3K9me3 in human and 6 are enriched for H3K9me3 in chimpanzee. We found that the 256 KRAB-ZNF genes are no more likely to be differentially expressed between human and chimpanzee than all 17,354 genes (52% of KRAB-ZNFs are classified as differentially expressed and 49% of all genes; FDR of 1%; *Figure 6—figure supplement 9A* and Table 11 in *Supplementary file 1*). There are 14 KRAB-ZNF genes whose TSS overlaps an H3K9me3 region that is enriched in human (eight) or chimpanzee (six). The association between silencing and gene expression divergence is consistent with the pattern we observed for all genes whose TSS overlaps an H3K9me3 region (*Figure 6—figure supplement 4*) but due to the small numbers in each category, we cannot reject the null hypothesis of no association (*Figure 6—figure supplement 9B*).

Finally, as H3K9me3 is not the only repressive histone modification in the genome, it is possible that we do not observe a strong association between silencing and expression divergence because there is redundancy in histone-based TE silencing mechanisms. To address this possibility, we considered genome-wide data from the Polycomb-repressed associated histone modification H3K27me3 in human H1 hESCs. We found that TEs preferentially silenced by H3K9me3 in chimpanzee are not associated more often with increased levels of H3K27me3 compared to TEs preferentially silenced by H3K9me3 in human (*Figure 6—figure supplement 10*). This suggests that H3K27me3 and H3K9me3 are not redundant in silencing TEs across species.

## Discussion

The potential for TEs to participate in the regulation of gene expression has received increasing attention as genomic technologies to study these elements advance (*Davidson and Britten, 1979*; *Jordan et al., 2003*; *Feschotte, 2008*; *Göke and Ng, 2016*). However, the role of TEs can be paradoxical. They can serve as regulatory sequence when they are bound by transcription factors, or localize in regions of open chromatin. Conversely, TEs can be actively silenced to prevent unwanted activity in a cell-type dependent manner. One cell type where TE activity can be particularly

deleterious to genome integrity is embryonic stem cells. Indeed, it was found that TE activity is generally restricted through repressive histone modifications in mouse embryonic stem cells (*Schlesinger and Goff, 2015*; *Robbez-Masson and Rowe, 2015*). TE activity and regulation has been studied in many contexts, but there are no published comparative studies of TE silencing in primates.

In order to gain insight into how TE silencing mediated by the TRIM28/SETDB1 pathway evolves in pluripotent cells from the hominid lineage, we profiled the distribution of the repressive histone modification H3K9me3 in human and chimpanzee iPSCs. The rationale behind our approach is the expectation that TEs that are marked by H3K9me3 are silent and are therefore not mobile, nor able to act as regulatory elements in this cell type. In contrast, TEs that are unmarked and not silenced, could be expressed, or capable of transposition, or participation in gene regulation. Alternatively, these elements may have acquired sufficient mutations to lose the ability to transpose, or influence gene regulatory programs, and are therefore not silenced because they do not 'threaten' genome integrity. We were particularly interested in finding differences in silencing of the same TEs between species, as this could imply that these TEs act as regulatory elements in only one species, and could contribute to phenotypic differences between species.

We found that there are differences in the TE families that are preferentially silenced in both species. This suggests specificity of silencing and targeting mechanisms, which are shared in humans and chimpanzees. In our study, up to 60% of SVA, 50% of ERV, and 15% of L1 families overlap with H3K9me3 in both species. It is likely that TE silencing in our human and chimpanzee iPSC lines is mediated by the TRIM28 co-repressor as it is known to be involved in silencing ERVs in mouse ESCs (*Rowe et al., 2010*), and SVAs, HERVs and L1 elements in human ESCs (*Turelli et al., 2014*; *Castro-Diaz et al., 2014*). This specificity is likely achieved through sequence-specific KRAB-ZNFs that recruit TRIM28. Indeed, specific KRAB-ZNFs recognize specific TE families (*Jacobs et al., 2014*; *Imbeault et al., 2017*; *Schmitges et al., 2016*).

While TEs silenced by H3K9me3 in both species tend to be less diverged from the corresponding TE consensus sequence than those that are not silenced, TEs that have large inter-species differences in silencing (more than twofold) tend to resemble TEs that are not silenced. This could imply that TEs whose silencing differs across species are less likely to require silencing, or that they are more likely to have enhancer function in just one species. Indeed, amongst TE-derived enhancers defined by CAGE in human, there is a depletion of younger, primate-specific TEs (*Simonti et al., 2017*). Assuming an evolutionary arms race model, one expects that TEs and their KRAB-ZNF repressors would co-evolve; indeed there is a correlation between the age of KRAB-ZNF genes and their target TEs (*Najafabadi et al., 2015*), and related KRAB-ZNFs bind related TEs (*Schmitges et al., 2016*). However, the recent observation that many KRAB-ZNF gene – TE pairs arose at the same time during evolutionary history, and that this relationship is maintained to this day despite the fact that these TEs are no longer able to transpose, challenges the idea of a broad evolutionary arms race (*Imbeault et al., 2017*). In our study, we found silencing by H3K9me3 in TEs that are no longer transposition competent. It could be that these TEs are not silenced to prevent their mobility, but rather to prevent their potential regulatory activity.

TEs that are silenced in both species are located further away from the TSS than TEs that are silenced preferentially in one species. This difference between the 'shared' and 'species-specific' silenced TEs indicates that we do not have an overwhelming proportion of false negatively classified TEs in the 'shared' group. This observation also raises the possibility that TEs that are silenced only in one species are more likely to contribute to gene expression differences between species. However, we did not find this to be the case. Within species, genes with silenced TEs that overlap promoters have lower expression levels, on average, than genes with proximal TEs that are not silenced. Across species, inter-species differences in TE silencing are only associated with divergence in gene expression when the TEs are located in promoter regions, within 1 kb of the TSS. Further from genes, we focused on TEs that overlap enhancer regions, as Imbeault *et al.*, have shown that KRAB-ZNF gene-bound TEs only affect neighboring gene expression if the TE overlaps marks of enhancers (*Imbeault et al., 2017*). In our data, however, we did not find such an effect.

Our observation of overall lack of association between TE silencing and expression divergence should be considered in the following context: First, while it is intuitive to relate a promoter to a gene, it is more challenging to relate a distant enhancer to the gene it regulates. Second, it has recently been shown that while many TEs contribute to primate-specific *cis*-regulatory elements

(CREs), many of these TEs act as transcriptional repressors and hence silencing of these elements may in fact lead to increased expression of neighboring genes (*Trizzino et al., 2017*). Third, we did not investigate the functional consequences of sequence changes to orthologous TEs, so our results do not account for the fact that mutations in the sequence of TEs in one species may inactivate them, eliminating the need for histone-based silencing. Fourth, we have restricted our analysis to orthologous genes; it is possible that regulatory novelty co-occurs with gene novelty, though admittedly this is rare when the species considered are human and chimpanzee. Fifth, we were unable to interrogate silencing of recently active TEs, due to our stringent H3K9me3 ChIP-seq read mapping strategy, which excludes reads that do not map uniquely.

Overall, we identified tens of thousands of inter-species differences in silencing at individual TE instances; however, considering that there are roughly four million human-chimpanzee orthologous TE sequences in the genome, these differences account for a minority of elements (2%). Of the TEs that are silenced, the observed level of conservation of TE silencing in human and chimpanzee is in line with what is observed for all H3K9me3 regions in the genome irrespective of the presence of a TE. Conservation of active regulatory regions, determined by H3K27ac, in human and chimpanzee is also thought to be similar (*Prescott et al., 2015*). It may be that even divergence at a small fraction of regulatory elements between human and chimpanzee – reflecting tens of thousands of instances – is sufficient to explain much of the observed divergence in gene expression levels. Yet, although we performed multiple analyses to uncover such a relationship, we concluded that, at most, inter-species differences in TE silencing have a modest effect on gene expression divergence in iPSCs.

We acknowledge that proving the absence of a species-specific effect is challenging. In general, it is difficult to draw strong conclusions based on the inability to reject a null hypothesis. Nevertheless, our observations are robust with respect to a wide range of statistical thresholds used to identify TE silencing or gene expression differences. Our observations should also be interpreted in the context of our initial assumption about what can be learned from characterizing TE silencing by H3K9me3 in iPSCs. First, while H3K9me3-mediated TE silencing is a well-established repressive mechanism in pluripotent cells, additional mechanisms of TE silencing exist; including RNA interference, CpG methylation and other repressive histone modifications such as H3K27me3 (*Slotkin and Martienssen, 2007*; *Huda et al., 2010*; *Day et al., 2010*; *Leeb et al., 2010*), and thus our findings may not extend to these other silencing pathways. For example, Marchetto *et al.*, showed that there is differential regulation of L1 elements by piRNA-associated PIWIL2 and APOBEC3B in human and chimpanzee iPSCs (*Marchetto et al., 2013*). Pluripotent stem cells can be considered to be an usual cell type in many respects, including TE regulation. In these cells, TE activity is known to be regulated by a histone-based silencing mechanism following the global resetting of DNA methylation. While there are examples of this TE repression pathway in differentiated cell types (*Ecco et al., 2016*; *Brattås et al., 2017*), it is largely believed that DNA methylation replaces histone-mediated repression following the DNA de-methylation phase in embryogenesis. Therefore, alternative TE silencing mechanisms predominant in other cell types, such as DNA methylation, may be more dynamic across species. Similarly, one should also consider that the ability of TEs to act as enhancer elements has been shown to be cell-type specific (*Xie et al., 2013*). Our results may therefore not extend to other somatic cell types, which are not under the same level of regulatory constraint as iPSCs.

Our results complement several previous studies, which explored the role of transposable elements in primate gene regulation. The primate-specific HERVH LTR element is a human pluripotency marker (*Wang et al., 2014*) and has been shown to have similar levels of expression in chimpanzee iPSCs (*Ramsay et al., 2017*). Indeed, we find HERVH to be expressed in human and chimpanzee iPSCs, and to be depleted for H3K9me3, suggesting that this element is essential for the pluripotency network in both humans and chimpanzees. Many open chromatin regions in human are derived from primate-specific TEs (*Jacques et al., 2013*), and newly evolved CREs are enriched in young SVA and LTR elements (*Trizzino et al., 2017*). Trizzino *et al.* found that 2% of all LTRs are active (marked by H3K27ac) in liver but a quarter of genes with an LTR insertion affect gene expression divergence between multiple primate species, and that less than 1% of CREs are differentially active between human and chimpanzee. This suggests that while many TEs, specifically young TEs, are important for influencing inter-species differences in expression, the majority of TEs in the genome do not act as regulatory sequences.

When considering the broader context for the contribution of mammalian TEs to mediating inter-species differences in gene regulation, previous elegant work has dissected the role of specific TEs in inter-species differences between mice and rats (*Chuong et al., 2013*; *Sundaram et al., 2017*). TE activity is substantial in the rodent lineage compared to hominids (*Lander et al., 2001*), which suggests that TEs can more quickly rewire regulatory networks in these species. Consistent with the reduction in TE activity in primates compared to other mammals, ChIP-seq assays for the highly conserved transcriptional insulator CTCF, have shown reduced TE-mediated propagation of CTCF binding sites in primates compared to other mammalian lineages (*Schmidt et al., 2012*; *Schwalie et al., 2013*).

In summary, to date, there have been few genome-wide studies demonstrating the contribution of TEs to gene regulation in primates. TEs have been shown to comprise most primate-specific regulatory sequence (*Jacques et al., 2013*), to contribute to primate-conserved non-coding transcripts (*Ramsay et al., 2017*), and lineage-specific gene regulation (*Trizzino et al., 2017*). More broadly, while gene expression differences between primates have been shown to often be associated with inter-species differences in histone modifications (*Cain et al., 2011*; *Zhou et al., 2014*), the chromatin landscape is generally highly conserved in primates. For example, as many as 84% of genomic regions enriched for the active histone modification, H3K27ac, are similar between humans and chimpanzees in iPSC-derived neural crest cells (*Prescott et al., 2015*). Our results provide further support for the notion that chromatin state is generally highly conserved between closely related primate species. We have not found evidence that TEs, whether orthologous or not, have substantially rewired pluripotent stem cell gene regulatory networks between humans and chimpanzees.

# Materials and methods

**Key resources table**

| Reagent type (species) or resource | Designation | Source or reference | Identifiers | Additional information |
|---|---|---|---|---|
| Cell line (*H. sapiens*, Female) | H20682 iPSC | This paper | | Age 23, Caucasian, fibroblast origin |
| Cell line (*H. sapiens*, Male) | H20961 iPSC | 10.1371/journal.pgen.1005793 | | |
| Cell line (*H. sapiens*, Female) | H21194 iPSC | 10.1371/journal.pgen.1005793 | | |
| Cell line (*H. sapiens*, Female) | H21792 iPSC | This paper | | Age 20, Caucasian, fibroblast origin |
| Cell line (*H. sapiens*, Male) | H28126 iPSC | 10.1371/journal.pgen.1005793 | | |
| Cell line (*H. sapiens*, Male) | H28815 iPSC | This paper | | Age 19, Caucasian, fibroblast origin |
| Cell line (*H. sapiens*, Female) | H18489 iPSC | 10.1101/gr.224436.117 | | |
| Cell line (*H. sapiens*, Female) | H18511 iPSC | 10.1101/gr.224436.117 | | |
| Cell line (*H. sapiens*, Male) | H19098 iPSC | 10.1101/gr.224436.117 | | |
| Cell line (*H. sapiens*, Male) | H19101 iPSC | 10.1101/gr.224436.117 | | |
| Cell line (*P. troglodytes*, Female) | C3647 iPSC | 10.7554/eLife.07103 | | |
| Cell line (*P. troglodytes*, Male) | C3649 iPSC | 10.7554/eLife.07103 | | |
| Cell line (*P. troglodytes*, Female) | C3651 iPSC | 10.7554/eLife.07103 | | |
| Cell line (*P. troglodytes*, Female) | C40210 iPSC | 10.7554/eLife.07103 | | |
| Cell line (*P. troglodytes*, Female) | C40280 iPSC | 10.7554/eLife.07103 | | |
| Cell line (*P. troglodytes*, Male) | C4955 iPSC | 10.7554/eLife.07103 | | |
| Cell line (*P. troglodytes*, Male) | C8861 iPSC | 10.7554/eLife.07103 | | |
| Antibody | anti-Histone H3K9me3 (rabbit polyclonal) | Abcam | abcam:8898 | 5 ug |

## Samples

We used 7 biological replicates (individuals) for chimpanzee and 10 biological replicates for human (encompassing two different populations). This number of replicates has been shown to be sufficient to capture intra-species variation allowing for robust identification of inter-species differences in gene expression (*Gallego Romero et al., 2015*; *Cain et al., 2011*; *Zhou et al., 2014*). Each biological replicate (i.e. different individual within a species) was assayed once. Technical replication was not performed. Experimental processing was designed such that technical variables (such as RNA extraction batch) were not confounded with the variable of interest (species).

## iPSC culture

All chimpanzee and Caucasian (CAU) iPSCs were reprogrammed from fibroblasts with episomal vectors (*Gallego Romero et al., 2015*; *Burrows et al., 2016*), while the Yoruba (YRI) iPSCs were similarly reprogrammed from lymphoblastoid cell lines (LCLs) (*Banovich et al., 20162018*). It has been shown, however, that cell-type of origin does not affect iPSC DNA methylation or gene expression patterns (*Burrows et al., 2016*). After reprogramming, iPSC colonies were cultured on a Mouse Embryonic Fibroblast feeder layer for 12–15 passages prior to conversion to feeder-free growth for 6–26 passages. Three new iPSC lines (H20682, H21792, H28815) were generated as previously described for H20961 (referred to as Ind1 F-iPSC in Burrows *et al.*), H28126 (Ind3 F-iPSC) and H21194 (Ind4 F-iPSC) (*Burrows et al., 2016*).

Human and chimpanzee feeder-independent stem cell lines were maintained at 70% confluence on Matrigel hESC-qualified Matrix (354277, Corning, Bedford, MA) at a 1:00 dilution. Cells were cultured in Essential 8 Medium (A1517001, ThermoFisher Scientific, Waltham, MA) at 37°C with 5% (vol/vol) $CO_2$ with daily media changes. Cells were passaged by enzyme-free dissociation (0.5 mM EDTA, 300 mM NaCl in PBS), and seeded with ROCK inhibitor Y-27632 (ab120129, Abcam, Cambridge, MA). All cell lines tested negative for mycoplasma contamination.

## iPSC quality control

All iPSCs were assessed for the expression of pluripotency markers, differentiation ability, and genomic instability: these cells express pluripotency factors, can spontaneously differentiate into all three germ layers following embryoid body formation, and display normal karyotypes. Gene expression from all samples was measured using the HumanHT12 Illumina Gene Expression Array, and data analyzed using the PluriTest bioinformatic assay to determine whether their global gene expression signature matches that of known pluripotent cells (*Müller et al., 2011*). We also assayed for the presence of episomal reprogramming vector sequence by RT-PCR. All three new iPSC lines (H20682, H21792, H28815), and previously described iPSC lines passed these quality control metrics (*Gallego Romero et al., 2015*; *Banovich et al., 20162018*; *Burrows et al., 2016*). We identified one human (H28815) that tested positive for episomal reprogramming vector sequence. However, because this sample was not an obvious outlier in our data we chose to include it in our study.

## Immunocytochemistry

iPSCs were fixed in 4% paraformaldehyde in PBS, permeabilized in PBS-T (0.25% Triton X-100 in PBS), and blocked for 30 min in 2.5% BSA in PBS-T. Cells were incubated with primary antibodies in 2.5% BSA in PBS-T overnight at a 1:100 dilution: SSEA-4 mouse IgG (sc-21704, Santa Cruz Biotechnology, Santa Cruz, CA), and Oct-3/4 rabbit IgG (sc-9081). Secondary antibodies were diluted to 1:500 in 2.5% BSA-PBS-T: donkey anti-mouse IgG, Alexa 488 (A21202, ThermoFisher Scientific), donkey anti-rabbit IgG Alexa 594 (A21207) and incubated for 1 hr at room temperature. Nuclei were counter-stained with 1 µg/ml Hoechst 33342 (H3570, ThermoFisher Scientific).

## ChIP-seq

ChIP-seq experiments were largely performed according to a published protocol (*Schmidt et al., 2009*). Briefly, 30 million cells were cross-linked with 1% formaldehyde for 10 min at room temperature followed by quenching with 2.5 M glycine. Following cell lysis and sonication (Covaris S2: 4 min, duty cycle 10%, five intensity, 200 cycles per burst in $4 \times 6 \times 16$ mm tubes per individual), lysates were incubated with 5 µg H3K9me3 antibody (8898, Abcam) overnight. ChIP and 50 ng input DNA from each individual was end-repaired, A-tailed and ligated to paired-end Illumina TruSeq ChIP

Sample Preparation kit sequencing adapters before 18 cycles of PCR amplification. 200–300 bp DNA fragments were selected for sequencing. ChIP enrichment at known target genes (*ZNF333* and *ZNF554*) was quantified and confirmed by qPCR. ChIP-seq experiments were performed in two species-balanced batches. Input and ChIP libraries were multiplexed separately and 50 base pairs sequenced paired-end on the HiSeq2500 in rapid run mode according to the manufacturer's instructions.

Sequencing data quality was verified using FastQC (http:// www.bioinformatics.babraham.ac.uk/ projects/fastqc/). Paired-end ChIP-seq reads from each species were mapped to their respective genome (hg19 or panTro3) using BWA (version 0.7.12) (*Li and Durbin, 2009*) and a mapping quality filter of MAQ > 10. PCR duplicates were removed by samtools (version 0.1.19)(*1000 Genome Project Data Processing Subgroup et al., 2009*). Properly-paired reads only were selected for further analysis.

ChIP and Input reads from each individual were used to call peaks in each individual using MACS2 (*Zhang et al., 2008*) with the *broadpeak* option at various *qvalue* cut-offs. A lenient threshold of *qvalue* = 0.1 was used in subsequent analysis to maximize the number of regions identified in each species.

To determine the minimum number of sequencing reads required to saturate the number of identified peaks, a read sub-sampling analysis was performed in two individuals from each species. This analysis revealed that the number of peaks called approaches saturation at a median read depth of 6.9 million paired ChIP reads across the four individuals tested (*Figure 1—figure supplement 7A*). Sub-sampling Input reads, while maintaining the number of ChIP reads constant, saturates the number of peaks called at a median of 5.7 million paired Input reads across the four samples (*Figure 1—figure supplement 7B*). The read depth of the vast majority of samples (31/34) fall within this range.

Peak co-ordinates in each individual in each species were then converted to the alternative genome using liftOver (*Speir et al., 2016*) and the best reciprocal chain (http://hgdownload.cse.ucsc.edu/goldenPath/hg19/vsPanTro3/reciprocalBest/) between hg19 and panTro3 requiring 70% sequence match. Peak regions in each individual that could be reciprocally and uniquely mapped between the two genomes were kept to generate a list of orthologous peak regions. Orthologous peak regions in each individual were concatenated to generate a final orthologous H3K9me3 region set across both species. 50-mer mappability for every base in the hg19 and panTro3 genome was determined based on the GEM algorithm (*Derrien et al., 2012*). Mappability of all orthologous regions was determined, and only those regions with an average mapping quality score > 0.8 in each species were retained.

The number of H3K9me3 ChIP-seq and Input reads falling into these orthologous peak regions in each individual from each species was determined using HTSeq (*Anders et al., 2015*). Filtered, autosomal H3K9me3 ChIP-seq and Input read counts in each orthologous region (peak) from each individual in each species were compared using spearman correlation analysis. Having used Input samples to identify regions of H3K9me3 enrichment and identify ChIP-seq samples with low signal, only H3K9me3 ChIP-seq samples were taken forward in the analysis. As sample H20961 was an outlier it was removed from further analysis. H3K9me3 counts were standardized and log transformed to generate log$_2$cpm values for PCA analysis.

## Identifying differentially enriched H3K9me3-enriched regions

Orthologous H3K9me3 regions with 0 H3K9me3 counts in more than half of the individuals (>8) were removed. The R/Bioconductor package DESeq2 (*Love et al., 2014*) was used to identify which of the orthologous H3K9me3 regions have differential enrichment in H3K9me3 read counts between humans and chimpanzees. Each individual within a species was considered to be a replicate sample. Regions with low mean read counts were filtered prior to multiple-testing correction. We controlled significance at FDR < 0.01 by the Benjamini-Hochberg method. Regions with a log$_2$ fold change > 0 and an adjusted p-value of < 0.01 were considered to be 'Human-enriched', and those with log$_2$ fold change < 0 and an adjusted p-value of < 0.01 as 'Chimpanzee-enriched'. Those regions not significantly differentially enriched are defined as 'Shared'.

Differential H3K9me3 enrichment across species was also assessed by transforming read counts by voom and identifying differentially enriched regions using limma (*Law et al., 2014*). Significance was controlled at FDR < 0.01.

The proportion of regions differentially enriched at a range of $\log_2$ fold changes was calculated both using DESeq2 output, and using an Empirical Bayes approach with adaptive shrinkage (ashr) (*Stephens, 2017*).

### Orthologous TE generation

The RepeatMasker track (Smit, AFA, Hubley, R and Green, P. *RepeatMasker Open-3.0.*1996–2010 http://www.repeatmasker.org)(*Jurka, 2000*) from the hg19 genome assembly (*Speir et al., 2016*) was converted to panTro3 genome coordinates using liftOver reciprocal chain files, requiring a 70% sequence match. These panTro3 coordinates were intersected with the panTro3 RepeatMasker track, requiring that 50% of base pairs overlap, and that the hg19-derived TE name matches that of the panTro3 RepeatMasker annotation. This set was subsequently lifted back over to hg19 again requiring a 70% match. The same procedure was followed starting with the panTro3 RepeatMasker track. Once RepeatMasker tracks were confidently obtained on the same genome, they were intersected (requiring 50% overlap). The distribution of orthologous TEs within the five main TE classes (LINE, SINE, LTR, DNA and SVA) is similar to the overall distribution of TE classes in each species (namely, when we do not require TEs to be orthologous; [*Figure 2—figure supplement 1*]).

### Identifying silenced TEs

To obtain a high confidence set of TEs that are silenced, TEs that overlap H3K9me3-enriched regions by at least 50% of their length are considered to be 'Overlapping'. We acknowledge that our stringent H3K9me3 region and orthologous TE definitions likely mean that young, active TEs will be excluded from the majority of our analyses.

### Non-orthologous TE analysis

The non-orthologous TE set for each species was generated after excluding the orthologous TEs from all RepeatMasker-annotated TEs in that species. The list of species-specific TEs was generated by selecting non-orthologous TEs, where the TE name is not present in the RepeatMasker track of the other species. When determining the overlap between H3K9me3 regions and TEs with differing orthology status (orthologous, non-orthologous, and species-specific), we considered all H3K9me3 regions identified in each species independently (regardless of orthology).

### TE sequence divergence from the consensus TE sequence

The number of substitutions between the sequence of each TE instance and the consensus TE sequence (milliDiv score), was obtained from the RepeatMasker track in human.

### Orthologous TSS annotation

A file containing annotated human transcription start sites (TSS) from hg19 was downloaded from the Table Browser of the UCSC genome browser (http://genome.ucsc.edu/cgi-bin/hgTables) by filtering for the 'txStart' from Ensembl genes (*Karolchik et al., 2004*). A single TSS was assigned per gene using the TSS of the 5' most transcript from genes oriented on the sense strand, and the 3' most transcript from genes on the anti-sense strand. Only TSSs corresponding to 30,030 genes contained in a primate orthologous exon file were retained (*Blekhman et al., 2010*).

### Chromatin states of orthologous TEs

Orthologous TEs categorized by H3K9me3 silencing category (Shared, Human-enriched, Chimpanzee-enriched and Non-overlapping) were intersected with hg19 chromatin states defined by ChromHMM (*Ernst and Kellis, 2010*) from data produced and analyzed by the ENCODE Consortium (*ENCODE Project Consortium, 2012*) and obtained from the UCSC Genome Browser hg19 database (http://hgdownload.cse.ucsc.edu/goldenPath/hg19/database/) (*Rosenbloom et al., 2013*). Chromatin states from H1 hESCs (wgEncodeBroadHmmH1hescHMM.bed), HepG2 (wgEncodeBroadHmmHepg2HMM.txt) and GM12878 (wgEncodeBroadHmmGm12878HMM.txt) were used. An intersection is defined as a 1 bp overlap.

## Associating orthologous TEs with functional genomics annotations

Orthologous TEs categorized by H3K9me3 silencing category (Shared, Human-enriched, Chimpanzee-enriched and Non-overlapping) were intersected with ChIP-seq and DNase I hypersensitivity data obtained from the H1 hESC line (http://hgdownload.cse.ucsc.edu/goldenPath/hg19/database/) as described above. An intersection is called when 50% of the length of the TE overlaps with the genomic feature.

The following ENCODE datasets from the UCSC Genome Browser hg19 database were used: wgEncodeBroadHistoneH1hescH3k4me1StdPk.txt

wgEncodeBroadHistoneH1hescH3k4me3StdPk.txt
wgEncodeBroadHistoneH1hescH3k27acStdPk.txt
wgEncodeBroadHistoneH1hescH3k27me3StdPk.txt
wgEncodeAwgDnaseUwdukeH1hescUniPk.txt
wgEncodeAwgTfbsHaibH1hescP300V0416102UniPk.txt
wgEncodeAwgTfbsHaibH1hescPol2V0416102UniPk.txt
wgEncodeAwgTfbsHaibH1hescCtcfsc5916V0416102UniPk.txt
wgEncodeAwgTfbsHaibH1hescYy1sc281V0416102UniPk.txt
wgEncodeAwgTfbsHaibH1hescNanogsc33759V0416102UniPk.txt
wgEncodeAwgTfbsHaibH1hescPou5f1sc9081V0416102UniPk.txt

TEs have been found to mediate the differential wiring of pluripotency networks in humans and mice by carrying species-specific transcription factor binding sites for the core pluripotency factors Nanog and Oct4 (*Kunarso et al., 2010*). To determine whether inter-species TE silencing differences at these TE-associated transcription factor binding sites could mediate inter-species differences in the regulation of pluripotency, we overlaid our orthologous TEs with binding locations for these transcription factors and two ubiquitously expressed factors (CTCF and YY1). We found less than 0.5% of TEs overlapping these transcription factor binding locations (<100 instances per category) making it difficult to make robust conclusions (*Figure 5—figure supplement 6*). However, the small number of instances indicates that differences in H3K9me3-mediated TE silencing do not contribute to rewiring pluripotency networks in these closely-related species.

## RNA-seq

RNA was extracted from ~3 million cells using the ZR-Duet DNA/RNA extraction kit (D7001, Zymo, Irvine, CA) in two species-balanced batches. All RNA samples were of good quality: RIN scores of 10 were obtained for all samples except C40280 (9.7). All 17 samples were pooled together for RNA-seq library generation using the Illumina Truseq kit. The library pool was sequenced 50 base pairs, paired-end on the Hiseq2500.

Paired-end RNA-seq reads from each species were trimmed and mapped to each genome using Tophat2 (version 2.0.13) (*Kim et al., 2013*). Only properly-paired reads were taken for further analysis. The number of sequencing reads is similar across individuals from both species (median human: 42,557,599, median chimpanzee: 33,987,431) (*Figure 6—figure supplement 1* and Table 3 in *Supplementary file 1*). The number of reads falling into orthologous meta-exons across 30,030 Ensembl genes from hg19, panTro3 and rheMac3 (*Blekhman et al., 2010*) was determined using featureCounts within subread (version 1.5.0) (*Liao et al., 2014*). Autosomal genes with > 0 counts in > 10/17 individuals were retained. Counts were quantile normalized across all samples, standardized, and log-transformed to generate gene expression levels expressed as normalized $\log_2$cpm.

To identify genes differentially expressed between the two species, counts were transformed by voom (*Law et al., 2014*) prior to differential expression analysis using limma (*Smyth, 2004*). We controlled significance at FDR < 0.01.

## Association between H3K9me3-mediated orthologous TE silencing and neighboring gene expression

We selected the annotated TSSs described above that correspond to orthologous genes for which we have gene expression data. We used the genomic coordinates of these annotations to define 1, 10, 20 and 40 kb windows upstream of the TSS. We then identified all TEs contained within each window, retaining the previous annotation of whether the TE overlaps an H3K9me3 region or not. Those TEs overlapping an orthologous H3K9me3 region had additional information in the form of

H3K9me3 counts in each individual of each species. H3K9me3 counts in each orthologous H3K9me3 region overlapping a TE were summed within a window to yield a final H3K9me3 TE silencing count per gene per species. If multiple TEs fell within the same orthologous H3K9me3 region, this region was only counted once. H3K9me3 count values overlapping all TEs within a window was used as input for DESeq2 to determine inter-species differences in TE silencing per gene as previously described.

To determine gene expression divergence of genes overlapping H3K9me3 regions at the TSS, regardless of the presence of a TE, a 1 kb region around the TSS was created. All 141,642 H3K9me3 orthologous regions (irrespective of the whether the regions overlaps a TE or not) were overlapped with the defined TSS corresponding to orthologous genes. Genes where at least 50% of the length of the 1 kb TSS region overlap with an H3K9me3 region are considered to be overlapping the TSS.

## Effect of TEs within human H3K4me1 and H3K4me3 regions on inter-species gene expression divergence

To identify TEs with potential to act as regulatory regions important in mediating inter-species gene expression differences, a subset of orthologous TEs that also overlap a human H3K4me1/H3K4me3 region in the H1 hESC line were used. These TEs were assigned their closest gene based on the location of the TSS, and only unique orthologous genes were retained. Inter-species gene expression divergence was determined for these genes and compared to the divergence of unique orthologous genes closest to all H3K4me1/H3K4me3 regions in H1 hESCs.

## TE expression

The expression of TE types was determined as previously described (*Karimi et al., 2011*). Briefly, multi-mapping, paired-end RNA-seq reads were retained for both humans and chimpanzees. RepeatMasker coordinates for all annotated instances of each TE type were obtained in each species. RPKM values for each TE type, in each species, were calculated by normalizing agglomerated read coverage of all annotated genomic copies of that TE type in the reference genome, to the agglomerated length of the TE type, as well as the total number of exonic reads. TE types were considered to be expressed if they had RPKM values > 1.

## Association between H3K9me3-mediated non-orthologous TE silencing and neighboring gene expression

In each species independently, we used all previously determined non-orthologous TEs annotated as overlapping or not overlapping any H3K9me3 region identified in that species. Non-orthologous TEs were overlapped with the 1, 10 and 20 kb windows identified upstream of each orthologous gene's TSS as described above. To identify TSS positions in the chimpanzee genome, the 1 bp TSS annotations of orthologous genes determined in human were lifted over to the chimpanzee genome as previously described in the generation of orthologous TEs. Similarly, a 1 kb region around each human TSS was lifted over to the chimpanzee genome, and only orthologous single bp TSSs where the broader 1 kb region lifts over to the chimpanzee genome were retained.

Each gene was categorized as 'silent' if all non-orthologous TEs in the window upstream of the TSS overlap H3K9me3, or 'non-silent' if at least one non-orthologous TE in the upstream window did not overlap with a H3K9me3 region. Results are consistent if we further stratify categories of genes where upstream TEs are fully silenced (all TEs overlap H3K9me3), or non-silenced (no TEs overlap H3K9me3).

## KRAB-ZNF gene expression and overlap with H3K9me3 regions

To identify those KRAB-ZNF genes marked by H3K9me3, a curated list of 346 genes (*Kapopoulou et al., 2016*) was filtered to include only those contained within our set of orthologous genes. These 256 genes were overlapped with orthologous H3K9me3 regions. KRAB-ZNF genes where 50% of their length overlaps with H3K9me3 are considered to be overlapping.

KRAB-ZNF genes where 50% of the length of a 1 kb region around the TSS overlaps with an orthologous H3K9me3 region were considered to be overlapping H3K9me3 at the TSS, and used in downstream gene expression divergence analysis.

## Data access
All data have been deposited in the Gene Expression Omnibus (www.ncbi.nlm.nih.gov/geo/) under accession number GSE96712. https://www.ncbi.nlm.nih.gov/geo/query/acc.cgi?token=sloxgksifvinn-wr&acc=GSE96710

## Acknowledgements

We thank Matthew Lorincz as well as all members of the Gilad lab, especially Sebastian Pott, for helpful and insightful discussions, and Amy Mitrano for performing RNA extractions and preparing sequencing libraries. We thank the ENCODE Consortium for making their data freely available to use. MCW is supported by an EMBO Long-Term Fellowship (ALTF 751–2014) and the European Commission Marie Curie Actions.

## Additional information

### Funding

| Funder | Grant reference number | Author |
| --- | --- | --- |
| National Institute of General Medical Sciences | GM077959 | Yoav Gilad |
| EMBO Long-Term Fellowship/ European Commission Marie Curie Actions | ALTF 751-2014 | Michelle C Ward |

The funders had no role in study design, data collection and interpretation, or the decision to submit the work for publication.

### Author contributions

Michelle C Ward, Conceptualization, Formal analysis, Investigation, Visualization, Methodology, Writing—original draft, Project administration, Writing—review and editing; Siming Zhao, Formal analysis; Kaixuan Luo, Mohammad M Karimi, Formal analysis, Writing—review and editing; Bryan J Pavlovic, Resources; Matthew Stephens, Formal analysis, Supervision, Writing—review and editing; Yoav Gilad, Conceptualization, Formal analysis, Supervision, Funding acquisition, Writing—original draft, Writing—review and editing

### Author ORCIDs

Michelle C Ward (iD) http://orcid.org/0000-0003-1485-320X
Bryan J Pavlovic (iD) http://orcid.org/0000-0002-7751-5315
Yoav Gilad (iD) http://orcid.org/0000-0001-8284-8926

### Decision letter and Author response

Decision letter https://doi.org/10.7554/eLife.33084.052
Author response https://doi.org/10.7554/eLife.33084.053

## Additional files

### Supplementary files
• Supplementary file 1. Supplemental Tables 1-11.
DOI: https://doi.org/10.7554/eLife.33084.043

• Supplementary file 2. Table with 141,642 orthologous H3K9me3 regions, their ChIP-seq read counts, DESeq2 results (baseMean, $log_2FC$, lfcSE, stat, pval, padj), and classifications as Shared, Human-enriched and Chimpanzee-enriched.
DOI: https://doi.org/10.7554/eLife.33084.044

• Supplementary file 3. Table with 4,242,188 orthologous TEs, their overlap with orthologous H3K9me3 regions, and classification as Shared, Human-enriched and Chimpanzee-enriched.

DOI: https://doi.org/10.7554/eLife.33084.045

• Transparent reporting form

DOI: https://doi.org/10.7554/eLife.33084.046

### Major datasets

The following dataset was generated:

| Author(s) | Year | Dataset title | Dataset URL | Database, license, and accessibility information |
|---|---|---|---|---|
| Ward MC | 2017 | Epigenomic conservation of transposable element silencing | https://www.ncbi.nlm.nih.gov/geo/query/acc.cgi?acc=GSE96712 | Publicly available at the NCBI Gene Expression Omnibus (accession no: GSE96712). |

The following previously published dataset was used:

| Author(s) | Year | Dataset title | Dataset URL | Database, license, and accessibility information |
|---|---|---|---|---|
| Rosenbloom | 2013 | ENCODE data in the UCSC Genome Browser | http://hgdownload.cse.ucsc.edu/goldenPath/hg19/database | Publicly available at the UCSC Genome Browser |

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
