## [Decision Letter]

Thank you for submitting your article "Silencing of transposable elements may not be a major driver of regulatory evolution in primate iPSCs" for consideration by *eLife*. Your article has been reviewed by three peer reviewers, and the evaluation has been overseen by a Reviewing Editor and Patricia Wittkopp as the Senior Editor. The following individuals involved in review of your submission have agreed to reveal their identity: Christopher Brown (Reviewer #2); Geoffrey Faulkner (Reviewer #3).

The reviewers have discussed the reviews with one another and the Reviewing Editor has drafted this decision to help you prepare a revised submission.

Summary:

This manuscript provides a comparative study of transposable element (TE) silencing in primates, profiling H3K9me3 using ChIP-seq in 17 induced pluripotent stem cell (iPSC) lines from humans and chimps. The authors compare these profiles across the individuals and species to identify sets of TEs that are consistently and divergently marked between species. The majority of orthologous TEs are consistently marked in both species. The authors interpret this as implying that there are few functional differences in TE silencing between humans and chimps. They further support this conclusion by analyzing RNA-seq data from the same cell lines. They find no association between differential silencing of TEs and the expression levels of nearby genes. They also demonstrate that shorter TEs, older TEs (based on distance to consensus), and TEs closer to gene starts are more likely to be differentially silenced. They further demonstrate that differentially silenced TEs not associated with large differences in gene expression.

The manuscript is well written, the experiments are well designed, the analyses are well conducted, and the manuscript's conclusions follow logically from the presented analyses. The findings will be of significance to those interested in cis-regulatory evolution, gene expression, and transposable element biology. However, there are aspects of the analyses that would benefit from further clarification. In addition, we all agreed that the work needs a more balanced presentation of conclusions, as well as discussion of experimental caveats and discussion of alternative conclusions.

Essential revisions:

1) A more balanced presentation of the Results and Discussion sections

Throughout the manuscript the levels of difference in H3K9me3 at orthologous TEs between humans and chimpanzees are characterized as remarkably conserved. These claims would be much stronger if the authors could provide a sense of how much difference in the H3K9me3 profiles would be required to count as meaningfully diverged. For example, the read count correlations between species are lower than within species (Results section), and 11% (16,238) of tested regions are differentially enriched. From one perspective, that's quite a bit of divergence, even if only a small fraction influences gene regulation. Indeed, 66 genes with species-specific H3K9me3 within 1 kb or the TSS show differential expression. Couldn't 66 such genes be functionally meaningful, especially given the similarity of humans and chimps?

How much differential methylation/expression is necessary to count as significant? Repeatedly, statistics are presented that could easily be interpreted the other way. For example, in subsection “Majority of orthologous TEs are similarly silenced in human and chimpanzee”: "Indeed, while ultimately most TEs (88%) do not overlap H3K9me3 regions, when TEs do overlap H3K9me3 regions, we are generally (for 82% of TEs) unable to find evidence that these TEs are silenced differently across species.” Equally one could read that as 18% of TEs (1000s of individual elements) being differentially silenced, right? This is a recurrent theme in the manuscript, that one could just as easily interpret this result the other way. Also, in subsection “Orthologous TEs tend to be silenced more often than species-specific TEs”: "We considered the ChIP-seq data in the context of the orthologous TEs, and observed that only 12% of orthologous TEs overlap an orthologous H3K9me3 region with at least 50% of their length" That's 12% of 4,248,188 (~500,000) right? That's a lot. Perhaps the issue here is an exclusive reliance on percentages rather than considering that the absolute number of elements involved is still very high, and that tends to disagree with the main message about TE silencing not being a major driver of regulatory innovation in primates.

2) Careful framing of the scope of the conclusions in the context of mechanisms which often silence TEs

The authors demonstrate that longer TEs are preferentially silenced. Why is this indicative of regulatory potential? What about being more likely to contain sequences that recruit silencing machinery? Have they performed a motif analysis or assessment of other genomic features? Are there features besides length, age, and distance to TSS that are associated with silencing of TEs? Open chromatin (available ATAC seq data), active histone mods (available), local TE density? Doesn't the fact that TEs that are further from the TSS are more likely to be silenced argue against the 'regulatory potential' hypothesis?

There are other means by which TEs are silenced. As the authors note, mutations to TE sequences commonly disrupt their activity and ability to influence transcription. Without accounting for sequence-level differences, it is challenging to interpret differences in the activity of orthologous TEs between species. For example, an orthologous TE may be inactive in both species while only marked in one due to an inactivating mutation in the other. This could account for the lack of correlation between H3K9me3 state and expression. Also, isn't it possible that the regulatory effect of the TE could be repressive, so silencing of the TE would not necessarily only lead to repression. Furthermore, as the authors note, these conclusions may not extend beyond the embryonic stem cell context.

Finally, the conclusion that TEs do not contribute to the regulatory divergence between species (e.g., in the Discussion section and other areas) is too strong given the data presented. The authors show that differences in H3K9me3 near the TSSes of genes do not strongly correlate with differences in expression between species. There are many other ways TEs could influence expression. In many cases, TEs have enhancer activity and differences in the silencing of enhancer TEs that influence expression would not be detected here given the focus on TSS proximal TEs. The authors could use enhancer maps to perform similar analyses of differentially silenced enhancer TEs or make clear that their analyses do not address this potential regulatory mechanism.

3) Address and clarify the technical concerns about the methods raised surrounding power, peak calling, TSS focus, and definition of orthologous peaks

For example, have the authors used their input data as a covariate in their differential silencing analyses? It is unclear as written. As written, it is unclear if peaks called in individuals are merged prior to differential expression analysis. If no merging was performed, are all DESEQ tests independent? The minimum read count threshold seems very low. Have the authors demonstrated that there is not substantial loss in power to detect differential histone modification at this low end?

Does this approach (Results section): "We defined orthologous H3K9me3 ChIP-seq regions as those where a ChIP-seq peak, contained within orthologous human-chimpanzee genomic regions, was identified in at least one individual, in either species in our study (see Methods)." exclude any regions that are polymorphic within chimp or human (I assume so).

The authors should note the caveat that using H3K9me3 as a mark of repression, whilst being reasonable, is not uniformly indicative of repression and, if other markers were used instead, some of the discordant TEs in chimp versus human may be concordant for these other markers.

Re: the use of sequence divergence from consensus scaled by mutation rate as an estimate of TE age. There are many factors that could confound this analysis, including differences in the mutation rate of different TE families and differences in mutation rate over different evolutionary epochs. If this metric is to be called age, I would want to see more benchmarking of its accuracy. How variable in this statistic for members of the same TE family and type? How does this accord with what we know about their periods of activity? Furthermore, Dfam provides estimates of the origin of all TE models based on their presence across different clades. Do the results hold when stratifying by TE origin? Otherwise, I would recommend reporting these results in terms of sequence divergence from consensus rather than age.

4) Better framing of these results in the context of somewhat contradictory studies

In the Discussion section, the authors should note more explicitly that their conclusions appear to contradict those made by several other studies in this area in recent years. For example, Wang et al., 2014 finds that LTRs are an essential part of pluripotency regulation in primates. Cordaux and Batzer, 2009 makes similar conclusions about L1. Jacques, Jeyakani and Bourque, 2013 suggests TEs are actually very important for regulatory innovation in primates. Considering how long the manuscript is, the authors should dedicate a section to, in a balanced and fair way, address why their findings are at odds with these other papers. I also do not think that anecdotal examples should be dismissed (e.g. " Overall, while studies have shown the effect of TE derived enhancers on gene expression divergence at single loci (81), or tens of loci (23), genome wide effects on global gene expression divergence have been less clear"), as these single locus examples can be important and instructive.

5) (Optional) The authors clearly demonstrate that most TEs are not silenced by H3K9me3 and that most silenced TEs are equivalently silenced in both species. I think it would be particularly interesting to ask a complementary set of questions: Are differentially silenced loci enriched for TEs? For species-specific TE insertions?

---

## [Author Response]

Essential revisions:1) A more balanced presentation of the Results and Discussion sectionsThroughout the manuscript the levels of difference in H3K9me3 at orthologous TEs between humans and chimpanzees are characterized as remarkably conserved. These claims would be much stronger if the authors could provide a sense of how much difference in the H3K9me3 profiles would be required to count as meaningfully diverged. For example, the read count correlations between species are lower than within species (Results section), and 11% (16,238) of tested regions are differentially enriched. From one perspective, that's quite a bit of divergence, even if only a small fraction influences gene regulation. Indeed, 66 genes with species-specific H3K9me3 within 1 kb or the TSS show differential expression. Couldn't 66 such genes be functionally meaningful, especially given the similarity of humans and chimps?

We agree with this comment. It is important for us to note that this is more a matter of perspective than flawed interpretation. Our own perspective is influenced by the fact that in other comparative studies with similar (or often a bit smaller) sample size, 50% of orthologous genes can be classified as differentially expressed, and of those, 15-20% are also associated with epigenetic marker divergence between species (such as methylation and/or chromatin accessibility). Thus, given these numbers, 11% of differentially enriched TEs, and only 66 genes whose inter-species expression divergence can perhaps be explained by TEs, are relatively small numbers. That said, we accept the comment and we have softened the language in the main text.

Regarding power, we agree – that really becomes the most important topic when the main observation is that we are most often unable to reject the null. The effect sizes of the differences that we do classify are rather small, and studies of other histone modifications with similar sample sizes have identified a much larger fraction of diverged loci. We realize that this is not a direct answer to the question ‘how much difference would be required?’ but it is not obvious how to answer that directly. We discuss power considerations, and more specifically the challenge of studies that are unable to reject the null, in the main text.

How much differential methylation/expression is necessary to count as significant? Repeatedly, statistics are presented that could easily be interpreted the other way. For example, in subsection “Majority of orthologous TEs are similarly silenced in human and chimpanzee”: "Indeed, while ultimately most TEs (88%) do not overlap H3K9me3 regions, when TEs do overlap H3K9me3 regions, we are generally (for 82% of TEs) unable to find evidence that these TEs are silenced differently across species.” Equally one could read that as 18% of TEs (1000s of individual elements) being differentially silenced, right? This is a recurrent theme in the manuscript, that one could just as easily interpret this result the other way. Also, in subsection “Orthologous TEs tend to be silenced more often than species-specific TEs”: "We considered the ChIP-seq data in the context of the orthologous TEs, and observed that only 12% of orthologous TEs overlap an orthologous H3K9me3 region with at least 50% of their length" That's 12% of 4,248,188 (~500,000) right? That's a lot. Perhaps the issue here is an exclusive reliance on percentages rather than considering that the absolute number of elements involved is still very high, and that tends to disagree with the main message about TE silencing not being a major driver of regulatory innovation in primates.

We agree – it is again a matter of perspective. It may be helpful to understand the perspective from which the study was conceived to get a sense of our prior expectation. This project was inspired by our earlier work that showed that while genes on human chromosome 21 are similarly regulated between humans and a Down syndrome mouse model carrying a copy of human chromosome 21, human-specific TEs become aberrantly activated (marked by H3K4me3) on human chromosome 21 in the mouse (Ward et al., 2013). We speculated that this was due to the fact that the host repressive factors needed to properly silence these elements were absent in the mouse nuclear environment ~60 million years diverged from human. This suggested a rapid evolution of TE silencing mechanisms in line with the evolutionary arms race hypothesis.

We now explain this perspective more clearly in the Introduction and we present both percentages and actual number of TE instances to put this better into perspective – subsection “Orthologous TEs tend to be silenced more often than species-specific TEs” as well as other instances in the manuscript e.g. subsection “The majority of orthologous TEs are similarly silenced in human and chimpanzee”.

2) Careful framing of the scope of the conclusions in the context of mechanisms which often silence TEsThe authors demonstrate that longer TEs are preferentially silenced. Why is this indicative of regulatory potential? What about being more likely to contain sequences that recruit silencing machinery? Have they performed a motif analysis or assessment of other genomic features? Are there features besides length, age, and distance to TSS that are associated with silencing of TEs? Open chromatin (available ATAC seq data), active histone mods (available), local TE density? Doesn't the fact that TEs that are further from the TSS are more likely to be silenced argue against the 'regulatory potential' hypothesis?

We don’t think that ‘having regulatory potential’ and being ‘more likely to contain sequences that recruit silencing machinery’ are mutually exclusive (we make that explicitly clear in the text now). We reason that if old TEs had regulatory potential the host would have developed mechanisms to silence younger instances of these elements. Indeed, citation 73 shows that within certain TE families, elements bound by multiple KRAB-ZNFs (known TF mediators of TE silencing) are more likely to be longer. There seems to be concordance with our data in terms of the families affected. TEs silenced in both species suggests to us that these elements are robustly silenced and likely to be located within regions of heterochromatin, whereas elements that show species-specific silencing, or no silencing, are closer to the TSS suggesting that these elements may have regulatory potential. We have clarified the rationale behind our perspective in subsection “Silenced TEs tend to have similar functional properties across species”.

That said, the reviewers raise a good point about actually testing the potential for TEs to act as regulatory sequence and determining whether there are differences between TE silencing categories. We have added three extra pieces of analysis to address this. Firstly, we overlaid our TEs categorized by inter-species silencing status with chromatin states in human embryonic stem cells as defined by ChromHMM and show that (i) the majority of TEs are associated with heterochromatin/low signal and (ii) that TEs that are preferentially silenced in human overlap annotated regulatory regions less frequently compared with TEs that are preferentially silenced in chimpanzee. This analysis is presented in Figure 5 and Figure 5—figure supplement 5 and is described in subsection “Silenced TEs tend to have similar functional properties across species”.

To gain further insight into the regulatory landscape associated with TEs in different silencing categories we overlaid our TEs with ChIP-seq and DNase hypersensitivity data in human embryonic stem cells. We found that a relatively small fraction of TEs overlap regions of open chromatin or ChIP-seq peaks: Less than 20% of TEs overlap active histone marks and less than 1% of TEs overlap DHS. We found that TEs that are preferentially silenced in human overlap regions of active gene regulation less often than TEs preferentially silenced in chimpanzee. The active chromatin annotations we considered include marking by H3K27ac, DHS, enhancer elements (marked by H3K4me1, p300) and regions marked by RNA polymerase II occupancy. Notably, there is no significant difference between TEs preferentially silenced in either species with respect to overlap with the active promoter-associated mark, H3K4me3. This analysis is presented in Figure 5—figure supplement 6 and described in subsection “Silenced TEs tend to have similar functional properties across species”.

As H3K9me3 is not the only repressive histone modification in the genome, we wanted to determine whether there is redundancy in TE silencing mechanisms. To address this, we overlapped our orthologous TEs with genome-wide data from the Polycomb-repressed associated histone modification H3K27me3 in human hESCs. We found that TEs that have less H3K9me3 in human are not more likely to be associated with increased levels of H3K27me3. This is presented in Figure 5—figure supplement 10 and is described in subsection “TE silencing is not a major driver of inter-species gene expression divergence”.

Finally, we determined the impact of TEs associated with enhancer and promoter histone marks on inter-species gene expression divergence. This analysis measures TEs’ association with gene expression slightly differently to what we did before (closest gene to individual TEs rather than all TEs in a window upstream of genes) and essentially results in the same pattern. We only see an effect of species-specific TE silencing on inter-species gene expression divergence when TEs overlap the promoter-associated mark, H3K4me3, which is in line with our previous analysis that showed that only TEs within 1 kb upstream of the TSS have a marginal effect. We don't see an effect when using TEs associated with the enhancer mark, H3K4me1, or in > 1 kb windows upstream of genes. This is shown in Figure 6—figure supplement 6 and is described in subsection “TE silencing is not a major driver of inter-species gene expression divergence”.

There are other means by which TEs are silenced. As the authors note, mutations to TE sequences commonly disrupt their activity and ability to influence transcription. Without accounting for sequence-level differences, it is challenging to interpret differences in the activity of orthologous TEs between species. For example, an orthologous TE may be inactive in both species while only marked in one due to an inactivating mutation in the other. This could account for the lack of correlation between H3K9me3 state and expression. Also, isn't it possible that the regulatory effect of the TE could be repressive, so silencing of the TE would not necessarily only lead to repression. Furthermore, as the authors note, these conclusions may not extend beyond the embryonic stem cell context.

The reviewers raise some valid scenarios that could affect the interpretation of the results. We have added these possible explanations to the Discussion section.

Finally, the conclusion that TEs do not contribute to the regulatory divergence between species (e.g., in the Discussion section and other areas) is too strong given the data presented. The authors show that differences in H3K9me3 near the TSSes of genes do not strongly correlate with differences in expression between species. There are many other ways TEs could influence expression. In many cases, TEs have enhancer activity and differences in the silencing of enhancer TEs that influence expression would not be detected here given the focus on TSS proximal TEs. The authors could use enhancer maps to perform similar analyses of differentially silenced enhancer TEs or make clear that their analyses do not address this potential regulatory mechanism.

We have noted in the discussion that while it is intuitive to relate a promoter to a gene, it is more challenging to relate a distal enhancer to the gene it regulates in the Discussion section. However, as described above, when overlaying TEs with enhancer marks we also do not see an effect on gene expression divergence.

3) Address and clarify the technical concerns about the methods raised surrounding power, peak calling, TSS focus, and definition of orthologous peaksFor example, have the authors used their input data as a covariate in their differential silencing analyses? It is unclear as written. As written, it is unclear if peaks called in individuals are merged prior to differential expression analysis. If no merging was performed, are all DESEQ tests independent? The minimum read count threshold seems very low. Have the authors demonstrated that there is not substantial loss in power to detect differential histone modification at this low end?

The input data was not used in the differential enrichment analysis as per the explicit recommendation of Dr. Michael Love, the author of the DESeq2 package. The input was only used to identify regions of enrichment in MACS2. This has been clarified in the Materials and methods section.

Peaks were called in each individual and merged to generate H3K9me3 regions. Each individual of each species was used independently as a replicate in the DESeq2 analysis. This has been clarified in subsection “ChIP-seq”.

We presented two read filtering cutoffs in the original manuscript (based on a minimum number of individuals within/between species with > 0 counts) as described above. The DESeq2 framework does not require pre-filtering as independent filtering occurs within the model. This is why the total number of tested regions is 141,642 not 150,390. A pre-filtering step increases computational efficiency. However, following the reviewer’s suggestion, we have now included an additional analysis based on a minimum threshold of 2 reads in >8 individuals and show that the percentage of differentially enriched regions is similar to less stringent pre-filtering (14% vs 11%). The results of this analysis have been added as a new table (Table 4 in Supplementary file 1) and is referenced in the Results section.

Does this approach (Results section): "We defined orthologous H3K9me3 ChIP-seq regions as those where a ChIP-seq peak, contained within orthologous human-chimpanzee genomic regions, was identified in at least one individual, in either species in our study (see Methods)." exclude any regions that are polymorphic within chimp or human (I assume so).

Orthologous regions were based on the human and chimpanzee reference genomes (independent of TE annotation). If H3K9me3 ChIP-seq reads passed the stringent mapping filtering within and between species, they were retained. As such, any H3K9me3 reads overlapping recently active TEs such as L1HS and AluYa elements, which may lead to different germline insertions between individuals within a species, would likely not be retained. A sentence about this category of elements, which we are unable to interrogate, has been added to the Discussion section. It is also mentioned in the Materials and methods section.

The authors should note the caveat that using H3K9me3 as a mark of repression, whilst being reasonable, is not uniformly indicative of repression and, if other markers were used instead, some of the discordant TEs in chimp versus human may be concordant for these other markers.

We acknowledge that there are additional silencing mechanisms in the genome but decided to focus this study on the output of the KRAB-ZNF-SETDB1 pathway that is prevalent in mouse embryonic stem cell TE silencing and is thought to contribute to the TE-host repressor evolutionary arms race hypothesis.

However, we have now overlaid our data with human H3K27me3 ChIP-seq data in human ESCs as described above. This analysis does not suggest that there is redundancy in TE silencing mechanisms, and hence cannot explain why no effects on inter-species gene expression are observed.

We have also been more explicit in the text about the silencing mechanisms this study does and does not address.

Re: the use of sequence divergence from consensus scaled by mutation rate as an estimate of TE age. There are many factors that could confound this analysis, including differences in the mutation rate of different TE families and differences in mutation rate over different evolutionary epochs. If this metric is to be called age, I would want to see more benchmarking of its accuracy. How variable in this statistic for members of the same TE family and type? How does this accord with what we know about their periods of activity? Furthermore, Dfam provides estimates of the origin of all TE models based on their presence across different clades. Do the results hold when stratifying by TE origin? Otherwise, I would recommend reporting these results in terms of sequence divergence from consensus rather than age.

While our approach to calculate repeat age by dividing sequence divergence by mutation rate has been used before (Bourque, *et al*., 2010; Ward*/Wilson*, *et al*., 2013), to eliminate the effects of potential mutation rate differences across TEs/TE families we now report our data as ‘divergence from the consensus sequence’ in the text in subsection “Silenced TEs tend to have similar functional properties across species” and in the Discussion section, Figure 5 and Figure 5—figure supplement 3.

4) Better framing of these results in the context of somewhat contradictory studiesIn the Discussion section, the authors should note more explicitly that their conclusions appear to contradict those made by several other studies in this area in recent years. For example, Wang et al., 2014 finds that LTRs are an essential part of pluripotency regulation in primates. Cordaux and Batzer, 2009 makes similar conclusions about L1. Jacques, Jeyakani and Bourque, 2013 suggests TEs are actually very important for regulatory innovation in primates. Considering how long the manuscript is, the authors should dedicate a section to, in a balanced and fair way, address why their findings are at odds with these other papers. I also do not think that anecdotal examples should be dismissed (e.g. " Overall, while studies have shown the effect of TE derived enhancers on gene expression divergence at single loci (81), or tens of loci (23), genome wide effects on global gene expression divergence have been less clear"), as these single locus examples can be important and instructive.

We have re-written much of the discussion taking into account all comments. We have elaborated on the particular studies mentioned and discussed how our study fits in. In most cases our study poses a slightly different question to previous studies and as such complements, rather than contradicts, earlier work.

Briefly, in accordance with Chuong, Elde and Feschotte, 2017 we show that primate-specific HERVH elements are expressed and similarly regulated in both human and chimpanzees, which suggests that this element is important not just in human but in at least two primate species. Marchetto et al., 2013 shows that there is differential L1 regulation between human and chimpanzee iPSCs and that this is due to a piRNA-mediated silencing pathway suggesting that different TE regulatory pathways may have different levels of cross-species conservation. Kazazian, 2004 looks at an active histone modification in primary liver tissue across multiple primate species and shows an enrichment of TEs in newly-evolved CREs. We have looked at a different cell type and histone modification in two closely-related primate species and therefore our findings do not necessarily contradict this study. Trizzino et al., 2017 show that 2% of all LTRs are active (marked by H3K27ac) in liver but a quarter of genes with an LTR insertion affect gene expression divergence between species, and that less than 1% of CREs are differentially active between human and chimpanzee. This suggests that while many TEs, specifically young TEs, are important for influencing inter-species differences in expression, potentially with large effects, the majority of TEs in the genome do not act as regulatory sequence.

We did not intend to be dismissive of single loci studies – we only meant to emphasize the difference between single loci and genome-wide studies. We have now revised the tone of the text and have recognized the importance of these studies.

5) (Optional) The authors clearly demonstrate that most TEs are not silenced by H3K9me3 and that most silenced TEs are equivalently silenced in both species. I think it would be particularly interesting to ask a complementary set of questions: Are differentially silenced loci enriched for TEs? For species-specific TE insertions?

This is an interesting complementary analysis. Given that the degree in overall similarity in H3K9me3 between species is similar to the degree of shared TE silencing between species we don’t suspect that this would give vastly different results.